# Learning a Subspace of Policies for Online Adaptation in Reinforcement Learning

**Jean-Baptiste Gaya**[*†]       **Laure Soulier**[†]       **Ludovic Denoyer**[*‡]

## Abstract

Deep Reinforcement Learning (RL) is mainly studied in a setting where the training and the testing environments are similar. But in many practical applications, these environments may differ. For instance, in control systems, the robot(s) on which a policy is learned might differ from the robot(s) on which a policy will run. It can be caused by different internal factors (e.g., calibration issues, system attrition, defective modules) or also by external changes (e.g., weather conditions). There is a need to develop RL methods that generalize well to variations of the training conditions. In this article, we consider the simplest yet hard to tackle generalization setting where the test environment is unknown at train time, forcing the agent to adapt to the system's new dynamics. This online adaptation process can be computationally expensive (e.g., fine-tuning) and cannot rely on meta-RL techniques since there is just a single train environment. To do so, we propose an approach where we learn a subspace of policies within the parameter space. This subspace contains an infinite number of policies that are trained to solve the training environment while having different parameter values. As a consequence, two policies in that subspace process information differently and exhibit different behaviors when facing variations of the train environment. Our experiments[1] carried out over a large variety of benchmarks compare our approach with baselines, including diversity-based methods. In comparison, our approach is simple to tune, does not need any extra component (e.g., discriminator) and learns policies able to gather a high reward on unseen environments.

## 1 Introduction

In recent years, Deep Reinforcement Learning (RL) has succeeded at solving complex tasks, from defeating humans in board games (Silver et al., 2017) to complex control problems (Peng et al., 2017; Schulman et al., 2017). It relies on different learning algorithms (e.g., A2C - (Mnih et al., 2016), PPO - (Schulman et al., 2017)). These methods aim at discovering a policy that maximizes the expected (discounted) cumulative reward received by an agent given a particular environment. If existing techniques work quite well in the classical setting, considering that the environment at train time and the environment at test time are similar is unrealistic in many practical applications. As an example, when learning to drive a car, a student learns to drive using a particular car, and under specific weather conditions. But at test time, we expect the driver to be able to generalize to any new car, new roads, and new weather conditions. It is critical to consider the generalization issue where one of the challenges is to learn a policy that generalizes and adapts itself to **unseen environments**.

Different techniques have been proposed in the literature (Section 6) to automatically adapt the learned policy to the test environment. In the very large majority of works, the model has access to multiple training environments (meta-RL setting). Therefore, the training algorithm can identify which variations (or invariants) may occur at test time and how to adapt quickly to similar variations. But this setting may still be unrealistic for concrete applications: for instance, it supposes that the student will learn to drive on multiple cars before getting their driving license.

---

[*]Facebook AI Research, correspondence to jbgaya@fb.com

[†]CNRS-ISIR, Sorbonne University, Paris, France

[‡]Now at Ubisoft

[1]Code available at `https://github.com/facebookresearch/salina/tree/main/salina_examples/rl/subspace_of_policies`

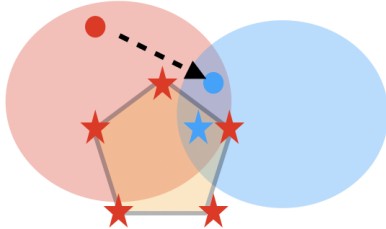 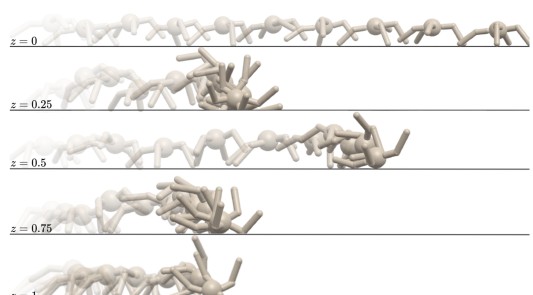

(a) The figure represents the parameter space. The red (resp. blue) region is the space of good policies over the training (resp. testing) environment. A single learned policy (red point) may be inefficient for the test environment and has to be adapted (e.g., fine-tuning) to become good at test-time (blue point). Instead of learning a single policy, we learn a convex sub-space (the pentagon) delimited by anchor policies (red stars) that aims at capturing a large set of good policies. Then the adaptation is just made by sampling policies in this subspace, keeping the best one (blue star).

(b) Qualitative example of k-shot adaptation on a modified Ant environment (20% of observations masked). 5 policies (i.e 5 values of $z$) are tested on one episode. In this case, for $z = 0.$, the Ant is able to adapt to this new environment. More example of LoP trajectories in Figures 5 and 6. See https://sites.google.com/view/subspace-of-policies/home for videos of the learned behaviors.

Figure 1: (Left) An illustration of the process of learning a subspace of policies. (Right) Comparison between PPO and our model in a test environment.

In this paper, we address a simpler yet harder to tackle generalization setting in which the learning algorithm is trained over one single environment and has to perform well on test environments; preventing us from using meta-RL approaches. A natural way to attack this setting is to start by learning a single policy using any RL algorithm, and to fine-tune this training policy at test time, over the test environment (See red/blue points in Figure 1a), but this process may be costly in terms of environment interactions.

Very recently, the idea of learning a set of diverse yet effective policies (Kumar et al., 2020b; Osa et al., 2021) has emerged as a way to deal with this adaptation setting. The intuition is that, if instead of learning one single policy, one learns a set of 'diverse' policies, then there is a chance that at least one of these policies will perform well over a new dynamics. The adaptation in that case just consists in selecting the best policy in that set by evaluating each policy over few episodes (K-shot adaptation). But the way this set of policies is built and the notion of diversity proposed in these methods have a few drawbacks: these models increase diversity by using an additional intrinsic reward which encourages the different policies to generate different distributions of states. This objective potentially favors the learning of policies that are sub-optimal at train time. Moreover, these approaches make use of an additional component in the policy architecture (e.g., a discriminator) that may be difficult to tune, particularly considering that, at train time, we do not have access to any test environment and thus cannot rely on validation techniques to tune the extra architecture.

Inspired by recent research on mode connectivity (Benton et al., 2021; Kuditipudi et al., 2019) and by (Wortsman et al., 2021) which aims to learn a subspace of models in the supervised learning setting, we propose to learn a **subspace of policies** in the parameter space as a solution to the online adaptation in the RL setting (see Figure 1a). Each particular point in this subspace corresponds to specific parameter values, and thus to a particular policy. This subspace is learned by adapting a classical RL algorithm (PPO and A2C in our case, see Section 3.3) such that an infinite continuum of policies is learned, each policy having different parameters. The policies thus capture and process information differently, and react differently to variations of the training environment (see Figure 1b). We validate our approach (Section 5) over a large set of reinforcement learning environments and compare it with other existing approaches. These experiments show that our method is competitive, achieves good results and does not require the use of any additional component of hyper-parameters tuning contrarily to baselines.

## 2  SETTING

**Reinforcement Learning:**    Let us define a state space $\mathcal{S}$ and an action space $\mathcal{A}$. In the RL setting, one has access to a training *Markov Decision Process* (MDP) denoted $\mathcal{M}$ defined by a transition distribution $P(s'|s,a) : \mathcal{S} \times \mathcal{A} \times \mathcal{S} \to \mathbb{R}+$, an initial state distribution $P^{(i)}(s) : \mathcal{S} \to \mathbb{R}+$ and a reward function $r(s,a) : \mathcal{S} \times \mathcal{A} \to \mathbb{R}$.

A policy is defined as $\pi_\theta(a|s) : \mathcal{S} \times \mathcal{A} \to \mathbb{R}+$, where $\theta$ denotes the parameters of the policy. A trajectory sampled by a policy $\pi_\theta$ given a MDP $\mathcal{M}$ is denoted $\tau \sim \pi_\theta(\mathcal{M})$. The objective of an RL algorithm is to find a policy that maximizes the expected cumulative (discounted) reward:

$$\theta^* = \arg \max_\theta \mathbb{E}_{\tau \sim \pi_\theta(\mathcal{M})}[R(\tau)] \tag{1}$$

where $R(\tau)$ is the discounted cumulative reward over trajectory $\tau$.

**Online adaptation:**    We consider the setting where the policy trained over $\mathcal{M}$ will be used over another MDP (denoted $\bar{\mathcal{M}}$) that shares the same state and action space as $\mathcal{M}$, but with a different dynamics and/or initial state distribution and/or reward function[2]. Importantly, $\bar{\mathcal{M}}$ is unknown at train time, and cannot be used for model selection, making the tuning of hyper-parameters difficult.

Given a trained model, we consider the K-shot adaptation setting where the test phase is decomposed into two stages: a first phase in which the model adapts itself to the new test environment over $K$ episodes, and a second phase in which the adapted model is used to collect the reward. We thus expect the first phase to be as short as possible (few episodes), corresponding to a fast adaptation to the new environment. Let us consider that a model $\pi_\theta$ generates a sequence of trajectories $\bar{\tau}_1, \bar{\tau}_2, ...., \bar{\tau}_{+\infty}$ over $\bar{\mathcal{M}}$, the performance of such a model, is defined as:

$$Perf(\pi_\theta, \bar{\mathcal{M}}, K) = \lim_{T \to \infty} \frac{1}{T} \sum_{t=1}^{T} R(\bar{\tau}_{K+t}) \tag{2}$$

which corresponds to the average performance of the policy $\pi_\theta$ over $\bar{\mathcal{M}}$ after $K$ episodes used for adapting the policy. Note that we are interested in methods that adapt quickly to new a test environment and we will consider small values of $K$ in our experiments. In the following, for sake of simplicity, $K$ will refer to the number of policies evaluated during adaptation since each policy may be evaluated over more than a single episode when facing stochastic environments.

## 3  LEARNING SUBSPACES OF POLICIES

**Motivation and Idea:**    To illustrate our idea, let us consider a toy example where the train environment contains states with correlated and redundant features, in such a way that multiple subsets of state features can be used to compute good actions to execute. Traditional RL algorithms will discover one policy $\pi_{\theta^*}$ that is optimal w.r.t the environment. This policy will typically use the state features in a particular way to decide the optimal action at each step. If some features become noisy (at test time) while, unluckily, $\pi_{\theta^*}$ particularly relies on these noisy features, the performance of the policy will drastically drop. Now, let us consider that, instead of learning just one optimal policy, we also learn a second optimal policy $\pi_{\theta^{*'}}$, but enforcing $\theta^{*'}$ to be different than $\theta^*$. This second policy may tend to make use of various features to compute actions. We thus obtain two policies instead of one, and we have more chances that at least one of these policies is efficient at test time. Identifying which of these two policies is the best for the test environment (i.e., adaptation) can simply be done by evaluating each policy over few episodes, keeping the best one. Our model is built on top of this intuition, extending this example to an infinite set of policies and to variable environment dynamics.

Inspired by Wortsman et al. (2021) proposing to learn a subspace of models for supervised learning, we study the approach of **learning a subspace of policies in the parameter space**, and the use of such a model for online adaptation in reinforcement learning. Studying the structure of the parameter space has seen a recent surge of interest through the *mode connectivity* concept (Benton et al., 2021; Kuditipudi et al., 2019; Wortsman et al., 2021) and obtain good results in generalization, but it has never been involved in the RL setting. As motivated in the previous paragraph, we expect that, given a variation of the training environment, having access to a subspace of policies that process information differently instead of a single policy will facilitate the adaptation. As a result, our method is very simple, does not need any extra hyper-parameter tuning and achieves good performance.

---

[2]In the experimental study, one training environment is associated to multiple test environments to analyze the ability to adapt to different variations.

## 3.1 SUBSPACES OF POLICIES

Given $\Theta$ the space of all possible parameters, a subspace of policies is a subset $\bar{\Theta} \subset \Theta$ that defines a set of corresponding policies $\bar{\Pi} = \{\pi_\theta\}_{\theta \in \bar{\Theta}}$.

Since our objective is to learn such a subspace, we have to rely on a parametric definition of such a subspace and consider $\bar{\Theta}$ as a simplex in $\Theta$. Let us define $N$ anchor parameter values $\bar{\theta}_1, .... \bar{\theta}_N \in \Theta$. We define the $\mathcal{Z}$-space as the set of possible weighted sum of the anchor parameters: $\mathcal{Z} = \left\{ z = (z_1, ...z_N) \in [0, 1]^N \mid \sum z_i = 1 \right\}$. The subspace we aim to learn is defined by:

$$\bar{\Theta} = \{\sum_{k=1}^{N} z_k \bar{\theta}_k, \forall z \in \mathcal{Z}\} \tag{3}$$

In other words, we aim to learn a convex hull of $N$ vertices in $\Theta$. Note that policies in this subspace can be obtained by sampling $z \sim p(z)$ uniformly over $\mathcal{Z}$.

The advantages of this approach are: a) the number of parameters of the model can be controlled by choosing the number $N$ of anchor parameters, b) since policies are sharing parameters (instead of learning a set of independent policies), we can expect that the learning will be sample efficient. Such a subspace is illustrated in Figure 1a through the "pentagon" (i.e., $N = 5$) in which angles correspond to the anchor parameters and the surface corresponds to all the policies in the built subspace.

**K-shot adaptation:** Given a subspace of policies $\bar{\Theta}$, different methods can be achieved to find the best policy over the test environment. For instance, it could be done by optimizing the distribution $p(z)$ at test time. In this article, we use the same yet effective K-shot adaptation technique than Kumar et al. (2020b) and Osa et al. (2021): we sample $K$ episodes using different policies defined by different values of $z$ that are uniformly spread over $\mathcal{Z}$. In our example, it means that we evaluate policies uniformly distributed within the pentagon to identify a good test policy (blue star). Note that, when the environment is deterministic, only one episode per value of $z$ needs to be executed to find the best policy, which leads to a very fast adaptation.

## 3.2 LEARNING ALGORITHM

Learning a subspace of policies can be done by considering the RL learning problem as maximizing:

$$\mathcal{L}(\bar{\Theta}) = \int_{\theta \in \bar{\Theta}} \mathbb{E}_{\tau \sim \pi_\theta}[R(\tau)] d\theta \tag{4}$$

Considering that $\bar{\Theta}$ is a convex hull as defined in Equation 3, and using the uniform distribution $p(z)$ over $\mathcal{Z}$, the loss function of Equation 4 can be rewritten as:

$$\mathcal{L}(\bar{\theta}_1, .... \bar{\theta}_N) = \mathbb{E}_{z \sim p(z)}\left[\mathbb{E}_{\tau \sim \pi_\theta}[R(\tau)]\right] \text{ with } \theta = \sum_{k=1}^{N} z_k \bar{\theta}_k \tag{5}$$

Maximizing such an objective function over $\bar{\theta}_1, .... \bar{\theta}_N$ outputs a (uniform) distribution of policies trained to maximize the reward, all these policies sharing common parameters.

**Avoiding subspace collapse:** One possible effect when optimizing $\mathcal{L}(\bar{\theta}_1, .... \bar{\theta}_N)$ is to reach a solution where all $\theta_k$ values are similar. In that case, all the policies would have the same parameters value, and will thus all achieve the same performance at test-time. Since we want to encourage the policies to process information differently, and following Wortsman et al. (2021), we encourage the anchor policies to have different parameters. This is implemented through the use of a regularization term denoted $C(\bar{\theta}_1, .... \bar{\theta}_N)$ that measures how much anchor policies are similar in the parameter space. This auxiliary loss is defined as a pairwise loss between pairs of anchor parameters:

$$C(\bar{\theta}_1, .... \bar{\theta}_N) = \sum_{i \neq j} cosine^2(\theta_i, \theta_j) \tag{6}$$

The final optimization loss is then:

$$\mathcal{L}(\bar{\theta}_1, .... \bar{\theta}_N) = \mathbb{E}_{z \sim p(z)}\left[\mathbb{E}_{\tau \sim \pi_\theta}[R(\tau)]\right] - \beta \sum_{i \neq j} cosine^2(\bar{\theta}_i, \bar{\theta}_j) \text{ with } \theta = \sum_{k=1}^{N} z_k \bar{\theta}_k$$

where $\beta$ is an hyper-parameter (see Section 5 for a discussion abot the tuning of this term) that weights the auxiliary term.

---

**Initialize:** $\bar{\theta}_1, \bar{\theta}_2, \phi$ (Critic), $n$ batch size

**1 for** $k = 0, 1, 2...$ **do**

2      Sample $z_1, ..., z_n \sim \mathcal{U}_{[0,1]}$

3      Define $\theta_{z_i} \leftarrow z_i \bar{\theta}_1 + (1 - z_i)\bar{\theta}_2$

4      Sample trajectories $\{\tau_i\}_1^n$ using $\{\pi_{\theta_{z_i}}\}$

5      Update $\bar{\theta}_1$ and $\bar{\theta}_2$ to maximize: $\frac{1}{n}\sum_{i=1}^n \widehat{\mathcal{L}}_{PPO}\left(\theta_{z_i}\right) - \beta \, cosine^2\left(\bar{\theta}_1, \bar{\theta}_2\right)$

6      Up. $\phi$ to minimize: $\frac{1}{n}\sum_{i=1}^n \widehat{\mathcal{L}}_{MSE}\left(\phi, z_i\right)$

**7 end**

---

Figure 2: The adaptation of the PPO Algorithm with the LoP model. The different with the standard PPO algorithm is that: a) trajectories are sampled using multiple policies $\theta_{z_i}$ b) The policy loss is augmented with the auxiliary loss, and c) The critic takes the values $z_i$ as an input.

### 3.3 LINE OF POLICIES (LoP)

In the case of $N = 2$, the subspace of policies corresponds to a simple segment in the parameter space defined by $\bar{\theta}_1$ and $\bar{\theta}_2$ as extremities. $\bar{\theta}_1$ and $\bar{\theta}_2$ are combined through a scalar value $z \in [0; 1]$:

$$\theta = z\bar{\theta}_1 + (1 - z)\bar{\theta}_2 \tag{7}$$

Computationally, learning a line of policies[3] is similar to learning a single policy for which the number of parameters is doubled, making this particular case a good trade-off between expressivity and training speed. It corresponds to the following objective function:

$$\mathcal{L}(\bar{\theta}_1, \bar{\theta}_2) = \mathbb{E}_{z \sim \mathcal{U}[0;1]}\left[\mathbb{E}_{\tau \sim \pi_{z\bar{\theta}_1 + (1-z)\bar{\theta}_2}}[R(\tau)]\right] - cosine^2(\bar{\theta}_1, \bar{\theta}_2) \tag{8}$$

We provide in Algorithm 7 the adapted version of the clipped PPO algorithm (Schulman et al., 2017) for learning a subspace of policies. In comparison to the classical approach, the batch of trajectories is first acquired by multiple policies sampled following $p(z)$ (line 2-3). Then the PPO objective is optimized taking into account the policies used when sampling trajectories (line 4). At last, the critic is updated (line 5), taking as an input the $z$ value so that it can make robust estimations of the expected reward for all the policies in the subspace. Adapting off-policy algorithms would be similar. Additional details are provided in appendix. Note that, for environments with discrete actions, we have made the same adaptation based on the A2C algorithm since A2C has less hyper-parameters than PPO and is easier to tune, with similar results.

## 4 EXPERIMENTS

We perform experiments in 6 different environments. Implementations based on the SaLinA (Denoyer et al., 2021) library together with train and test environments will be released upon acceptance. For each environment, we consider one train environment on which we trained the different methods, and multiple variations of the training environment for evaluation resulting in 50 test environments in total. The details of all the environment configurations and detailed performance are given in Appendix B. Note that the complete experiments correspond to hundred of trained policies, and dozens of thousands of policy evaluations. For simple control environments (i.e., CartPole, Pendulum and AcroBot), we introduce few variations of the physics constant at test-time, for instance by varying the mass of the cart, the length of the pole. For complex control environments (i.e., HalfCheetah and Ant using the BRAX library (Freeman et al., 2021), we both use variations of the physics (e.g., gravity), variations of the agent shape (e.g., changing the size of the leg, or of the foot) and sensor alterations. At last, in MiniGrid and ProcGen we perform experiments where the agent is trained in one particular levels, but is evaluated in other levels (single levels on MiniGrid, and set of 10 levels in ProcGen). Note that ProcGen is a pixel-based environment where the architecture of the policy is much more complex than in control environments. Toy experiments on a simple Maze 2d are given in Appendix B.8 to show the nature of the policies learned by the different methods.

We compare our approach **LoP**[4] with different state-of-the-art methods: a) The **Single** approach is just a single policy learned on the train environment, and evaluated on the test ones. b) The **DI-**

---

[3]Other ways to control the shape of the subspace can be used and we investigate some of them in Section 4

[4]We consider the LoP-A2C and the LoP-PPO models for environments with respectively discrete and continuous actions. LoP-PPO could be also used in the discrete case but requires more hyper-parameter tuning.

**AYN+R**(reward) method is an extension of DIAYN (Eysenbach et al., 2018) where a set of discrete policies is learned using a weighted sum between the DIAYN reward and the task reward:

$$R_{DIAYN+R}(s,a) = r(s,a) + \beta \log p(z|s) \tag{9}$$

Critically, this model requires to choose a discriminator architecture to compute $\log p(z|s)$ and modifies the train reward by defining an intrinsic reward that may drastically change the behavior of the policies at train time. c) At last, we also compare with the model proposed in (Osa et al., 2021) denoted **Lc** (Latent-conditioned) that works only for continuous actions. This model is also based on a continuous $z$ variable sampled uniformly at train time, but only uses an auxiliary loss without changing the reward. This auxiliary loss is defined through the joint learning of a density estimation model $\log P(z|s,a)$ where back-propagation is made over the action $a$. As in DIAYN+R, this model needs to carefully define a good neural network architecture for density estimation. Since Lc cannot be used with environment that have discrete actions, we have adapted DIAYN+R (called **DIAYN+R Cont.**) using a continuous $z$ variable (instead of a discrete one) and a density estimation model $\log P(z|s)$ as in Osa et al. (2021). Note that we do not compare to (Kumar et al., 2020a) for the exact same reason as the one identified in (Osa et al., 2021): SMERL assumes that the reward is known over the complete trajectories which results in unnatural adaptation of on-policy RL algorithms like PPO. Moreover, preliminary experiments with SMERL does not demonstrate any advantage against DIAYN+R correctly tuned. We also provide (see Table 4 in Appendix B.1) results where $K$ independent policies are learned, the best one being selected over each test environment. This approach obtains lower performance than the proposed baseline and needs $K$ more training samples making it unrealistic in most of the environments.

As network architectures, we use multi-layer perceptrons (MLP) with ReLU units for both the policy and the critic (detailed neural network architectures are described in Appendix). For DIAYN+R $\log P(z|s,...)$ is also modeled by a MLP with a soft-max output. For Lc and DIAYN+R Cont., $\log P(z|s,...)$ is modeled by a MLP that computes the mean of a Gaussian distribution with a fixed variance. For these baselines, $z$ is concatenated with the environment observation as an input for the policy and the critic models.

To choose the hyper-parameters of the different methods, let us remind that test environments cannot be used at train time for doing hyper-parameters search and/or model selection which makes this setting particularly difficult. Therefore, we rely on the following procedure: a grid-search over hyper-parameters is made, learning a single policy over the train environment. The best value of the hyper-parameters is then selected as the one that provides the best policy at train time. These hyper-parameters are then used for all the different baselines. Concerning the $\beta$ value, for LoP, we report test results for $\beta = 1.0$ while, for Lc and DIAYN+R, we use the best value of $\beta$ on test environments. This corresponds to an optimistic evaluation of the baseline performances; aiming at showing that our method is much more efficient since it does not need such a beta-tuning ($\beta = 1.0$ giving good performance in the majority of cases). Said otherwise, we compare our model in the less favorable case where baselines have been unrealistically tuned.

For the adaptation step, each policy is evaluated over 10 episodes for stochastic environments or 1 single episode for deterministic environments. We repeat this procedure over 10 different training seeds, and report the reward over the different test environments together with standard deviation. All detailed results are available in Appendix.

## 5 ANALYSIS

We report the test performance of the models on different environments in Table 1. In all the environments, the adaptive models perform better than learning a single policy over the train environment which is not surprising. In most of the cases, LoP is able to achieve a better performance than other methods. For instance, on HalfCheetah where we evaluate the different methods over 16 variations of the train environments, LoP achieves an average reward of 10589 while Lc and DIAYN+R obtain respectively 9547 and 9680 (standard deviations are reported in Appendix B). Some examples of the discovered that behaviors in Ant and HalfCheetah[5] for the different methods, and for different values of $z$ are illustrated in Figures 1b, 5 and 6. This outlines that learning models that are optimal on the train task reward, but with different parameter values, allows us to discover policies react differently to variations of the training environment. It seems to be a better approach than encouraging policies to have a different behaviors (i.e., generating different state distributions) at train time. Same

---

[5]Videos available at `https://sites.google.com/view/subspace-of-policies/home`

| | CartPole | Acrobot | Pendulum | Minigrid | Brax HalfCheetah | Brax Ant | ProcGen | toy maze |
|---|---|---|---|---|---|---|---|---|
| Nb. Test Env. | 6 | 6 | 3 | 6 | 16 | 15 | 3 | 4 |
| Type of actions | Discr. | Discr. | Discr. | Discr. | Cont. | Cont. | Discr. | Discr. |
| Single Policy | 143.4 | -99.7 | -52.7 | 0.169 | 7697 | 3338 | 11.09 | -83.2 |
| LoP | 149.9 | **-93.2** | **-28.9** | **0.447** | **10589** | **4031** | **16.38** | **-33.6** |
| DIAYN+R | **168.1** | -97.0 | -47.1 | 0.248 | 9680 | 3759 | 11.45 | -42.2 |
| DIAYN+R $L_2$ | 156.1 | -93.6 | -44.0 | 0.443 | - | - | - | - |
| Lc | - | - | - | - | 9547 | 4020 | - | - |

Table 1: Average cumulated reward of the different models over multiple testing environments averaged over 10 training seeds (higher is better). For DIAYN and Lc, we report the results and tested 10 policies for LoP, Lc and DIAYN+R using 10 episodes per policy for stochastic environments, and 1 episode per policy on deterministic ones. Performance is evaluated using the deterministic policy. Standard deviation is reported for each single test environment in Appendix B.

| | $K =$ | 5 | 10 | 20 | Train Perf. |
|---|---|---|---|---|---|
| LoP | $\beta = 0.1$ | 3905 | 3991 | 4164 | 7659 |
| | $\beta = 1.$ | **4035** | **4031** | **4174** | 7630 |
| | $\beta = 10$ | 3998 | 4012 | 4145 | 7670 |
| DIAYN | $\beta = 0.1$ | 3558 | 3833 | 3949 | 7739 |
| | $\beta = 1.$ | 3451 | 3759 | 2878 | 5388 |
| | $\beta = 10$ | 3356 | 3400 | 3109 | 4430 |
| Lc | $\beta = 0.1$ | 3909 | 4020 | 4150 | 7767 |
| | $\beta = 1.$ | 3820 | 3947 | 4126 | 7650 |
| | $\beta = 10$ | 3870 | 3945 | 4108 | 7710 |

| | | SmallFeet | TinyFriction | BigGravity |
|---|---|---|---|---|
| LoP | K=5 | 8283 | **10425** | **10464** |
| | K=10 | 8805 | **10662** | **10578** |
| | K=20 | 8794 | **10734** | **10807** |
| DIAYN+R | K=5 | 7580 | 9132 | 8989 |
| | K=10 | 7580 | 9132 | 8989 |
| | K=20 | 8255 | 10003 | 9766 |
| Lc | K=5 | 8186 | 9521 | 9360 |
| | K=10 | 8186 | 9661 | 9488 |
| | K=20 | 8107 | 9661 | 9506 |
| BoP (N=3) | K=5 | 6775 | 7867 | 7878 |
| | K=10 | 6660 | 7840 | 8026 |
| | K=20 | 6996 | 7963 | 8015 |
| CoP (N=3) | K=5 | **8996** | 9468 | 9287 |
| | K=10 | **9210** | 9523 | 9568 |
| | K=20 | **9155** | 9979 | 9695 |

Figure 3: (left) Performance of the models **at train time** that shows that for LoP $\beta$ is not hurting train performance while it is DIAYN+R. Standard error deviation is reported in Table B.2 for each environment. We also report the performace at train time that shows that a too high value of $\beta$ hurts DIAYN+R performance while is less critical in LoP. (right) Ablation study on the number of policies $K$ used at test time on 3 HalfCheetah environment variations (see Appendix B.1 for further details and additional results) together with the performance of the BoP and CoP variants. Standard deviation is given in appendix, Table 4.

conclusions can be drawn in most of the environments, including MiniGrid where LoP is able to explore large mazes while being trained only on small ones. On the ProcGen environment, where the observation is an image processed through a complex ConvNet architecture (See Appendix B.7), enforcing functional diversity (DIAYN+R) does not allow to learn good policies while the LoP model is able to better generalize to unseen levels. Note that the performance at train time is the same for all the different approaches reported (see the Table 3 for instance) but quickly decreases in DIAYN for larger values of $\beta$ while it stays stable for LoP where the best results are obtained for $\beta = 1.0$.

Interestingly, in CartPole, DIAYN+R performs quite well. Indeed, when analyzing the learned policies, it seems to be a specific case where it is possible to obtain optimal policies that are diverse w.r.t the states they are sampling (by moving the cart more or less on the right/left while maintaining the pole vertical).

We have also performed experiments where test environments have the same dynamics as the training environment, but with defective sensors (i.e., some features at test time have a null value – see Appendix B.2 Table 7 on the Ant environment). The fact that LoP behaves also well confirms the effectiveness of our approach to different types of variations, including noisy features on which baselines methods were not applied in previous publications.

**Sensitivity to hyper-parameters:** One important characteristic of LoP is that it can be used with $\beta = 1.0$ and does not need to define any classifier architecture as opposed to DIAYN+R and Lc. Indeed, as shown in Figure 3 (left), the training performance of DIAYN drastically depends on a good tuning of $\beta$. Lc, which is less sensible, needs to use a correct classifier architecture as in DIAYN. LoP is simple to tune since the cosine term is usually easy to satisfy and our approach, at convergence, always reaches a 0 value on this term when $\beta > 0.0$. As illustrated in Appendix B.1 and B2, it is also interesting to note than, on the BRAX environments, the number of environment interactions needed to train LoP is similar than the one needed to train a single policy and LoP comes with a very small overhead in comparison to classical methods.

**Online adaptation:** One interesting property is the number of policies (and thus of episodes) to test over a new environment to get a good performance. For LoP and Lc, given a trained model,

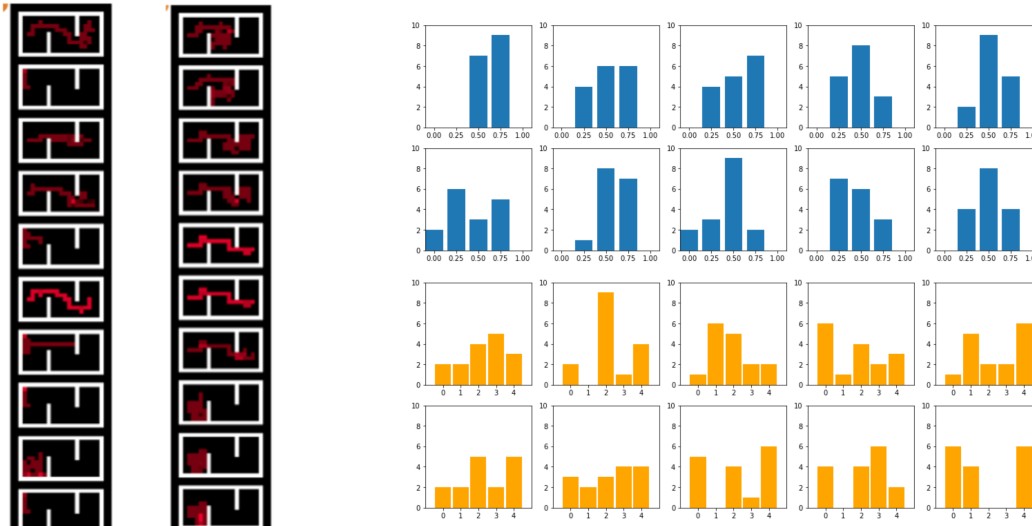

Figure 4: (left) Trajectories generated by $K = 10$ policies (one rows) on an unseen maze (objective is to go from left to right, see details in Appendix B.8) for DIAYN+R (left column with best $\beta$ value) and LoP (right column with $\beta = 1.0$). It illustrates the diversity obtained with DIAYN+R and LoP. (right) Number of times (y-axis) each policy (x-axis) (over $K = 5$ tested policies) is chosen over the 16 test environments of the HalfCheetah setting for each of the 10 seeds. Blue is LoP and orange is DIAYN+R. Different policies are used for different test environments showing the interest of learning a subspace of policies. Note that in LoP, the anchor policies are rarely chosen. Results for $K = 10$ and $K = 20$ in Appendix (Figure 7).

one can evaluate as many policies (i.e., different values of $z$) as desired. For DIAYN+R, testing more policies also means training more policies which is expensive and less flexible. Table 3 (right) provides the reward of the different methods when testing $K$ policies on different HalfCheetah settings: as expected, the performance of DIAYN+R tends to decrease when $K$ is large since the model has difficulties to learn too many diverse policies. For LoP and Lc, spending more episodes to evaluate more policies naturally leads to a better performance: these two models provide a better way to deal with the exploration-exploitation trade-off at test time. Again, please consider that Lc also needs to define an additional neural network architecture to model $\log P(z|s, a)$ while LoP does not, making our approach simpler.

**Beyond a Line of Policies:** While LoP is based on the learning of $N = 2$ anchor parameters, it is possible to combine more than two anchor parameters. We study two approaches combining $N = 3$ anchor parameters (that can be extended to $N = 3$): a) the first approach is a convex combination of policies (CoP) where $z$ is sampled following a Dirichlet distribution. (b) The second approach is a Bézier combination (BoP) as explained in Appendix A.2. The results are presented in Table 3 (right) over multiple HalfCheetah environments. It can be seen that these two strategies are not so efficient. LoP is thus a good trade-off between the number of parameters to train and the performance (Note that BoP and CoP need more samples to converge), at least given the particular neural network architectures we have used in this paper. We also performed an in-depth analysis of the evolution of the reward when K is increasing for LoP and CoP in Halfcheetah test environment (Figure 9 in Annex). While we expected CoP to outperform LoP when K is high, the best reward becomes stable when K=20 for both methods, and in most test environments, CoP is not able to reach the same best reward as LoP.

**Analysis of the learned policies:** To better understand the nature of the policies discovered by the different approaches, we have made a qualitative study in which we analyze i) the robustness of the methods to corrupted observations, ii) the functional diversity induced by the different models, and iii) the specificity of the different learned policies to particular test environments. First, LoP is more robust to input feature corruption (see Table B.2 for the results, and Table 5 for the setting in Appendix) and we conjecture that it is because the diversity in the parameter space allows this model to learn policies that does not take into account the same input features equally. We also measure the functional diversity induced by the different models by training *a posteriori* a classifier that

aims at recovering which policy (i.e which value of $z$) has generated particular trajectories (Exact protocol in Figure 8 in Appendix, with the training curves). On LoP with $K = 5$, such a classifier obtains a 82% accuracy at validation time showing that the 5 policies are quite diverse, but less than the DIAYN+R policies where the classifier reaches a 100% accuracy which is logical knowing the auxiliary loss introduced by DIAYN which enforces this type of diversity. It is interesting to note that with the trajectories generated in the test environments with LoP policies, the accuracy of the classifier is reaching 87 %: when LoP is facing new environments, it tends to generate more diverse policies. We think that it is due to the fact that, since the policies have different parameter values, they react differently to states that have not been encountered at train time. At last, Figure 4 (right) (and Figure 7 in appendix for K=10,20) illustrates which upon $K = 5$ policies is used for different test environments. It shows that both LoP and DIAYN+R use different policies over different test environments, showing that these methods are able to solve new environments by learning various policies and not a single but robust one. Examples of policies on a simple maze2d are given in Figure 4 (left) and Appendix which illustrate the diversity of the discovered policies.

## 6   RELATED WORK

Our contribution shares connections with different families of approaches. First of all, it focuses on the problem of online adaptation in Reinforcement Learning which has been studied under different terminologies: *Multi-task Reinforcement Learning* (Wilson et al., 2007; Teh et al., 2017), *Transfer Learning* (Taylor and Stone, 2009; Lazaric, 2012) and *Meta-Reinforcement Learning* (Finn et al., 2017; Hausman et al., 2018; Humplik et al., 2019). Many different methods have been proposed, but the best majority considers that the agent is trained over multiple environments such that it can identify variations (or invariant) at train time. For instance, Duan et al. (2016) assume that the agent can sample multiple episodes over the same environments and methods like (Kamienny et al., 2020; Liu et al., 2021) consider that the agent has access to a task identifier at train time.

More recently, diversity-based approaches have been adapted to focus on the setting where only one training environment is available. They share with our model the idea of learning multiple policies instead of a single one. For instance, DIAYN (Eysenbach et al., 2018) learns a discrete set of policies that can be reused and fine-tuned over new environments. It has been adapted to online adaptation in (Kumar et al., 2020a) where the authors propose to combine the intrinsic diversity reward together with the training task reward. This trade-off is obtained through a threshold-based method (instead of a simple weighted sum) with good results. But this method suffers from a major drawback identified in (Osa et al., 2021): it necessitates to sample complete episodes at each epoch which is painful and not adapted to all the RL learning algorithms. Osa et al. (2021) also proposed an alternative based on learning a continuous set of policies instead of a discrete one without using any intrinsic reward.

The method we propose is highly connected to recent researches on mode connectivity with neural networks. Mode connectivity is a set of approaches and analyses that focus on the shape of the parameter space. It has been used as a tool to study generalization in the supervised learning setting (Garipov et al., 2018), but also as a way to propose new algorithms in different settings (Mirzadeh et al., 2021). Obviously, the work that is the most connected to our approach is the model proposed in (Wortsman et al., 2021) that provides a way to learn a subspace of models in the supervised learning setting. Our contribution adapts this approach to RL for learning policies in a completely different setting which is online adaptation.

At last, our work is sharing similarities with robust RL which aims at discovering policies robust to variations of the training environment (Oikarinen et al., 2020; Zhang et al., 2021). The main difference is that robust RL techniques learn policies efficient 'in the worst case' and are not focused on the online adaptation to test environments (the objective is usually to learn a single policy efficient on different variations while we are learning multiple policies, just selecting one at test time).

## 7   CONCLUSION AND PERSPECTIVES

We investigate the idea of learning a subspace of policies in the reinforcement learning setting, and describe how this approach can be used for online adaptation. While simple, our method allows to obtain policies that are robust to variations of the training environments. Contrarily to other techniques, LoP does not need any particular tuning or definition of additional architectures to handle diversity, which is a critical aspect in the online adaptation setting where hyper-parameters tuning is impossible or at least very difficult. Future work includes the extension of this family of approaches in the continual reinforcement learning setting, the deeper understanding of the the built subspace and the investigation of different auxiliary losses to better control the shape of such a subspace.

## 8 REPRODUCIBILITY STATEMENT

We have made several efforts to ensure that the results provided in the paper are fully reproducible. In Appendix, we provide a full list of all hyperparameters and extra information needed to reproduce our experiments. The source code is available on SaLinA repository such that everyone can reproduce the experiments[6].

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

## A IMPLEMENTATION DETAILS

In this section, we provide details about the implementations of the baselines and our models. Appendix B provides details and additional results for each environment we used.

### A.1 LoP-PPO

We detail the losses used in algorithm 7. First, we recall the clipped-PPO surrogate objective described in Schulman et al. (2017). For a given trajectory $\tau = \{(s_t, a_t, r_t)\}_0^T$ collected by the policy $\pi_{\theta_{old}}$, and denoting $\rho_t(\theta) = \frac{\pi_\theta(a_t|s_t)}{\pi_{\theta_{old}}}$, the goal is to maximize:

$$\widehat{\mathcal{L}}_{PPO}(\theta) := \frac{1}{T} \sum_{t=0}^T \min \left[ \rho_t(\theta) A^{\pi_{\theta_{old}}}(s_t, a_t), clip(\rho_t(\theta), 1 + \epsilon, 1 - \epsilon) A^{\pi_{\theta_{old}}}(s_t, a_t) \right] \quad (10)$$

Where function $A^{\pi_{\theta_{old}}}$ is computed thanks to a value function $V_\phi$ by using Generalized Advantage Estimation (Schulman et al. (2018)). This function is simultaneously updated by regression on mean-squared error over the rewards-to-go $\widehat{R}_t$. In our case, this function not only takes $s_t$ as an input, but also the value $z$:

$$\widehat{\mathcal{L}}_{MSE}(\phi, z) := \frac{1}{T} \sum_{t=0}^T \left( V_\phi(s_t, z) - \widehat{R}_t \right)^2 \quad (11)$$

In HalfCheetah and Ant experiments, we sampled actions from a reparametrized Gaussian distribution using a squashing function (Ward et al. (2019)), but we set the standard deviation fixed (so it is an hyper-parameter encouraging exploration, called *action std* in Tables 3 and 6): the policy network only learns the mean of this distribution.

### A.2 BoP AND CoP

The only change between LoP and these models resides in the way we combine the $N$ anchor policies. For CoP, it is just the generalization of LoP for $N > 2$ (see 3.1). BoP, makes use of a Bezier parametric curve that uses Bernstein polynomials (the anchor parameters being the *control points*). For $N = 3$, it is defined by:

$$\bar{\Theta} = \left\{ (1 - z)^2 \bar{\theta}_1 + 2(1 - z) z \bar{\theta}_2 + z^2 \bar{\theta}_3, \quad \forall z \in [0, 1] \right\} \quad (12)$$

Concerning the policies $z$ evaluated at test time, BoP uses the same strategy as LoP by testing values that are uniformly distributed in $[0; 1]$. For CoP, we opted for sampling $K$ policies using a Dirichlet distribution over $[0, 1]^3$.

### A.3 DIAYN+R AND LC

In order to find the best trade-off between maximizing environment rewards and intrinsic rewards in DIAYN+R algorithm, we add the hyper-parameter $\beta$ :

$$R_{DIAYN+R}(s, a) = r(s, a) + \beta \cdot \log p(z|s) \quad (13)$$

As an alternative to DIAYN+R Osa et al. (2021) proposes an algorithm where the discriminator takes not only observations as an input but also the policy output, updating both discriminator $q_\phi$ and policy $\pi_\theta$ when back propagating the gradient. In this case, it is not necessary to add an intrinsic reward. While Osa et al. (2021) illustrate their methods with TD3 and SAC, we adapted it to PPO. The surrogate loss is given by:

$$\mathcal{L}_{LC} := \widehat{\mathcal{L}}_{PPO} + \beta \cdot \log q_\phi(z \mid s, \pi_\theta(.|s, z)) \quad (14)$$

# B  EXPERIMENTS DETAILS AND ADDITIONAL RESULTS

## B.1  HALFCHEETAH

Task originally coming from OpenAI Gym (Brockman et al., 2016). Instead of using MuJoCo engine, we decided to use Brax (Freeman et al., 2021) as it enables the possibility to acquire episodes on GPU. We use the vanilla environment for training.The policy and the critic are encoded by two different multi-layer perceptrons with ReLU activations. The base learning algorithm is PPO.

**Test environments:** we operated modifications similar as the ones proposed in (Henderson et al., 2017). Morphological variations: we changed the radius and mass of specific body parts (torso, thig, shin, foot). Variations in physics: we changed the gravity and friction coefficients.
Table 2 precisely indicates the nature of the changes for each environment.

| Env name | Modifications |
|---|---|
| BigFeet | Feet mass and radius $\times 1.25$ |
| BigFriction | Friction coefficient $\times 1.25$ |
| BigGravity | Gravity coefficient $\times 1.25$ |
| BigShins | Shins mass and radius $\times 1.25$ |
| BigThighs | Thighs mass and radius $\times 1.25$ |
| BigTorso | Torso mass and radius $\times 1.25$ |
| SmallFeet | Feet mass and radius $\times 0.75$ |
| SmallFriction | Friction coefficient $\times 0.75$ |
| SmallGravity | Gravity coefficient $\times 0.75$ |
| SmallShins | Shins mass and radius $\times 0.75$ |
| SmallThighs | Thighs mass and radius $\times 0.75$ |
| SmallTorso | Torso mass and radius $\times 0.75$ |
| HugeFriction | Friction coefficient $\times 1.5$ |
| HugeGravity | Gravity coefficient $\times 1.5$ |
| TinyFriction | Friction coefficient $\times 0.5$ |
| TinyGravity | Gravity coefficient $\times 0.5$ |

Table 2: Modified HalfCheetah environments used for testing. Morphological modifications include a variation on the mass and the radius of a specific part of the body (torso, thighs, shins, or feet). We also modified the dynamics (gravity and friction). Environment names are exhaustive: Big refers to a increase of 25% of radius and mass, Small refers to a decrease of 25%. For example, "BigFoot" refers to an HalfCheetah agent where feet have been increased in mass and radius by 25%. For gravity and friction, we also tried an increase/decrease by 50% (respectively tagged "Huge" and "Tiny").

| Hyper-parameter | Value |
|---|---|
| lr policy: | 0.0003 |
| lr critic: | 0.0003 |
| n parallel environments: | 2048 |
| n acquisition steps per epoch: | 20 |
| batch size: | 512 |
| num minibatches: | 32 |
| update epochs: | 8 |
| discount factor: | 0.99 |
| clip ration: | 0.3 |
| action std: | 0.5 |
| gae coefficient: | 0.96 |
| reward scaling: | 1. |
| gradient clipping: | 10. |
| n layers (policy): | 4 |
| n neurons per layer (policy): | 64 |
| n layers (critic): | 5 |
| n neurons per layer (critic): | 256 |
| LoP, BoP, CoP | |
| $\beta$: | 1 |
| DIAYN | |
| $\beta$: | 0.1 |
| lr discriminator: | 0.0001 |
| n layers (discriminator): | 2 |
| n neurons per layer (discriminator): | 64 |
| Lc | |
| $\beta$: | 10 |
| dimensions of z: | 1 |
| lr discriminator: | 0.001 |
| n layers (discriminator): | 2 |
| n neurons per layer (discriminator): | 64 |

Table 3: Hyper-parameters for PPO over HalfCheetah

| | **Single Policy** **(K=1)** | Ensemble **(5 policies)** | **LoP** | **DIAYN + R** | **Lc** | **BoP** | **CoP** |
|---|---|---|---|---|---|---|---|
| **K=5** | | | | | | | |
| BigFeet | 7433 ± 1988 | 8115 ± 344 | **8895 ± 289** | 8454 ± 655 | 8353 ± 804 | 7993 ± 702 | 8087 ± 496 |
| BigFriction | 8579 ± 2224 | 10757 ± 319 | **11635 ± 355** | 9962 ± 2113 | 10649 ± 2085 | 8846 ± 2279 | 10596 ± 1738 |
| BigGravity | 7508 ± 2086 | 9131 ± 260 | **10464 ± 798** | 8989 ± 1723 | 9360 ± 1531 | 7878 ± 1593 | 9287 ± 1200 |
| BigShins | 7274 ± 898 | 7992 ± 294 | **8879 ± 98** | 8099 ± 806 | 8206 ± 770 | 7726 ± 903 | 8226 ± 1144 |
| BigThig | 7963 ± 1466 | 9331 ± 310 | **10054 ± 724** | 8940 ± 1558 | 9360 ± 1669 | 8105 ± 1622 | 9525 ± 832 |
| BigTorso | 7091 ± 2221 | 8942 ± 94 | **9834 ± 310** | 8701 ± 1472 | 9198 ± 1342 | 7790 ± 1713 | 8927 ± 1128 |
| SmallFeet | 5973 ± 2490 | **10012 ± 1921** | 8283 ± 648 | 7580 ± 1411 | 8186 ± 1512 | 6775 ± 2308 | 8996 ± 1150 |
| SmallFriction | 8652 ± 1717 | 7299 ± 464 | **11391 ± 348** | 9813 ± 2074 | 10181 ± 1617 | 8652 ± 2075 | 10459 ± 1387 |
| SmallGravity | 9004 ± 1665 | 8004 ± 595 | **11840 ± 406** | 10132 ± 1903 | 10434 ± 1951 | 9428 ± 2180 | 10594 ± 1438 |
| SmallShin | 7492 ± 2999 | 10379 ± 257 | **10840 ± 196** | 9540 ± 1592 | 9837 ± 1343 | 8451 ± 1889 | 9967 ± 1079 |
| SmallThig | 8914 ± 1837 | 11133 ± 384 | **11294 ± 46** | 10078 ± 1410 | 10603 ± 1160 | 9340 ± 1751 | 10603 ± 1540 |
| SmallTorso | 8885 ± 1522 | 9669 ± 213 | **11433 ± 360** | 9850 ± 1541 | 10010 ± 1856 | 9182 ± 1779 | 10092 ± 1541 |
| HugeFriction | 6999 ± 3441 | 10006 ± 560 | **11537 ± 613** | 9483 ± 2228 | 10305 ± 2104 | 8387 ± 2515 | 10749 ± 1591 |
| HugeGravity | 6133 ± 2147 | **10452 ± 371** | 8425 ± 554 | 7700 ± 1216 | 7621 ± 1137 | 6532 ± 1133 | 7632 ± 903 |
| TinyFriction | 7953 ± 1843 | 9444 ± 293 | **10425 ± 211** | 9132 ± 1862 | 9521 ± 1462 | 7867 ± 2003 | 9468 ± 1468 |
| TinyGravity | 7304 ± 3484 | 9887 ± 1822 | **11385 ± 169** | 9363 ± 1734 | 9620 ± 2041 | 9118 ± 2360 | 9020 ± 2028 |
| **Average** | 7697 | 9410 | **10413** | 9114 | 9465 | 8254 | 9514 |

| | **Single Policy** **(K=1)** | **LoP** | **DIAYN + R** | **Lc** | **BoP** | **CoP** |
|---|---|---|---|---|---|---|
| **K=10** | | | | | | |
| BigFeet | 7433 ± 1988 | **8903 ± 246** | 8472 ± 535 | 8340 ± 888 | 8147 ± 835 | 8553 ± 1033 |
| BigFriction | 8579 ± 2224 | **11982 ± 330** | 10781 ± 1870 | 10705 ± 2092 | 9093 ± 2436 | 10850 ± 1526 |
| BigGravity | 7508 ± 2086 | **10578 ± 648** | 9766 ± 1424 | 9488 ± 1616 | 8026 ± 1809 | 9568 ± 803 |
| BigShins | 7274 ± 898 | **8854 ± 153** | 8276 ± 607 | 8340 ± 986 | 7831 ± 978 | 8753 ± 866 |
| BigThig | 7963 ± 1466 | **10335 ± 1001** | 9644 ± 1323 | 9427 ± 1532 | 8080 ± 1676 | 9591 ± 884 |
| BigTorso | 7091 ± 2221 | **10023 ± 427** | 9295 ± 1106 | 9271 ± 1317 | 7928 ± 1772 | 9131 ± 928 |
| SmallFeet | 5973 ± 2490 | 8805 ± 495 | 8255 ± 504 | 8186 ± 1458 | 6660 ± 1842 | **9210 ± 1063** |
| SmallFriction | 8652 ± 1717 | **11434 ± 358** | 10637 ± 1505 | 10204 ± 1663 | 8708 ± 2007 | 10459 ± 1387 |
| SmallGravity | 9004 ± 1665 | **11969 ± 341** | 10568 ± 1906 | 10444 ± 1984 | 9569 ± 2418 | 11334 ± 1429 |
| SmallShin | 7492 ± 2999 | **10764 ± 147** | 9990 ± 1107 | 9933 ± 1315 | 8441 ± 1801 | 9898 ± 982 |
| SmallThig | 8914 ± 1837 | **11524 ± 298** | 10689 ± 1137 | 10632 ± 1170 | 9430 ± 1881 | 10600 ± 1473 |
| SmallTorso | 8885 ± 1522 | **11567 ± 381** | 10328 ± 1736 | 10273 ± 1917 | 9228 ± 1736 | 10541 ± 1135 |
| HugeFriction | 6999 ± 3441 | **11659 ± 307** | 10335 ± 1955 | 10379 ± 2302 | 8526 ± 2710 | 10899 ± 1541 |
| HugeGravity | 6133 ± 2147 | **8793 ± 791** | 8124 ± 1086 | 7811 ± 1115 | 6573 ± 1138 | 8071 ± 352 |
| TinyFriction | 7953 ± 1843 | **10662 ± 385** | 10003 ± 1226 | 9661 ± 1497 | 7840 ± 2019 | 9523 ± 1537 |
| TinyGravity | 7304 ± 3484 | **11578 ± 460** | 9723 ± 2167 | 9650 ± 2228 | 9221 ± 2224 | 5107 ± 6910 |
| **Average** | 7697 | **10589** | 9680 | 9547 | 8331 | 9506 |

| | **Single Policy** **(K=1)** | **LoP** | **DIAYN + R** | **Lc** | **BoP** | **CoP** |
|---|---|---|---|---|---|---|
| **K=20** | | | | | | |
| BigFeet | 7433 ± 1988 | **9096 ± 257** | 8363 ± 800 | 8431 ± 803 | 8405 ± 961 | 8754 ± 1041 |
| BigFriction | 8579 ± 2224 | **12001 ± 276** | 9655 ± 2304 | 10779 ± 2158 | 9108 ± 2437 | 11124 ± 1397 |
| BigGravity | 7508 ± 2086 | **10807 ± 592** | 8695 ± 1550 | 9506 ± 1609 | 8015 ± 1834 | 9695 ± 879 |
| BigShins | 7274 ± 898 | **9036 ± 192** | 7922 ± 910 | 8393 ± 959 | 7994 ± 1046 | 8796 ± 415 |
| BigThig | 7963 ± 1466 | **10521 ± 1016** | 8639 ± 1315 | 9537 ± 1633 | 8159 ± 1602 | 9652 ± 1012 |
| BigTorso | 7091 ± 2221 | **10084 ± 449** | 8494 ± 1376 | 9383 ± 1363 | 7873 ± 1612 | 9462 ± 299 |
| SmallFeet | 5973 ± 2490 | 8794 ± 614 | 7540 ± 1155 | 8107 ± 1506 | 6996 ± 2483 | **9155 ± 497** |
| SmallFriction | 8652 ± 1717 | **11429 ± 319** | 9603 ± 2022 | 10271 ± 1680 | 8844 ± 2162 | 10498 ± 1331 |
| SmallGravity | 9004 ± 1665 | **12004 ± 259** | 9934 ± 2370 | 10521 ± 1986 | 9666 ± 2280 | 11382 ± 1377 |
| SmallShin | 7492 ± 2999 | **11041 ± 420** | 9135 ± 1715 | 10057 ± 1281 | 8709 ± 2127 | 10025 ± 803 |
| SmallThig | 8914 ± 1837 | **11571 ± 127** | 9970 ± 1847 | 10682 ± 1092 | 9563 ± 1820 | 10921 ± 1018 |
| SmallTorso | 8885 ± 1522 | **11673 ± 434** | 9620 ± 2009 | 10315 ± 1880 | 9301 ± 1814 | 10602 ± 1297 |
| HugeFriction | 6999 ± 3441 | **11724 ± 685** | 9438 ± 2471 | 10505 ± 2355 | 8553 ± 2603 | 11047 ± 1167 |
| HugeGravity | 6133 ± 2147 | **8930 ± 714** | 7498 ± 837 | 7897 ± 1147 | 6772 ± 1413 | 7925 ± 740 |
| TinyFriction | 7953 ± 1843 | **10734 ± 381** | 9001 ± 1908 | 9661 ± 1519 | 7963 ± 2234 | 9979 ± 878 |
| TinyGravity | 7304 ± 3484 | **11604 ± 388** | 9419 ± 2317 | 9822 ± 2200 | 9329 ± 2424 | 10822 ± 1045 |
| **Average** | 7697 | **10691** | 8933 | 9617 | 8453 | 9990 |

Table 4: Mean and standard deviation of cumulative reward achieved on HalfCheetah test sets per model (see Table 2 for environment details). Results are averaged over 10 training seeds (i.e., 10 models are trained with the same hyper-parameters and evaluated on the 16 test environments). $K$ is the number of policies tested at adaptation time, using 1 episode per policy since this environment is deterministic. Ensembling with $K = 5$ models takes 5 times more iterations to converge and testing values of $K > 5$ is very costly in terms of GPU consumption.

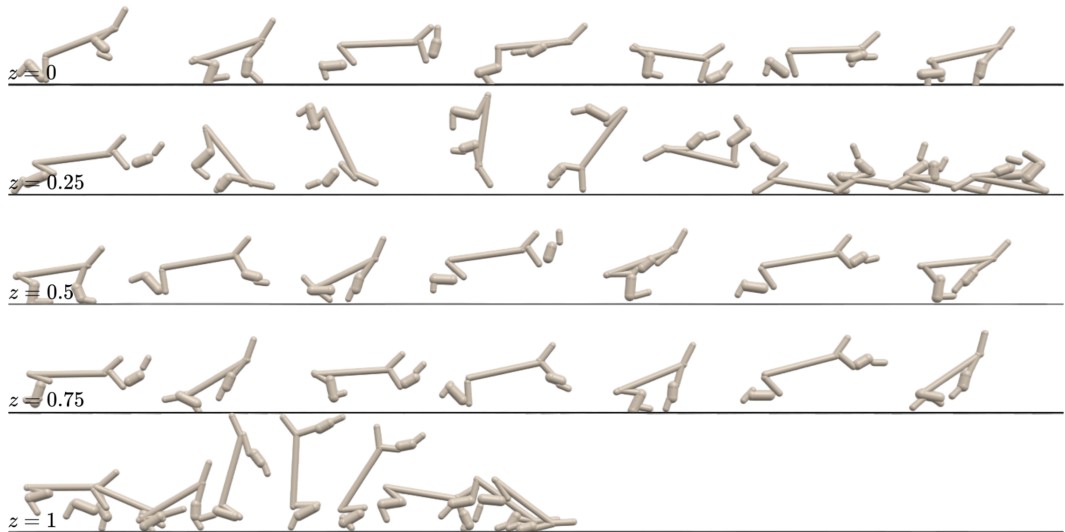

Figure 5: Qualitative example of LoP trajectories on HalfCheetah "BigShins" test environment (5-shot setting). The best reward is obtained for $z = 0.75$.

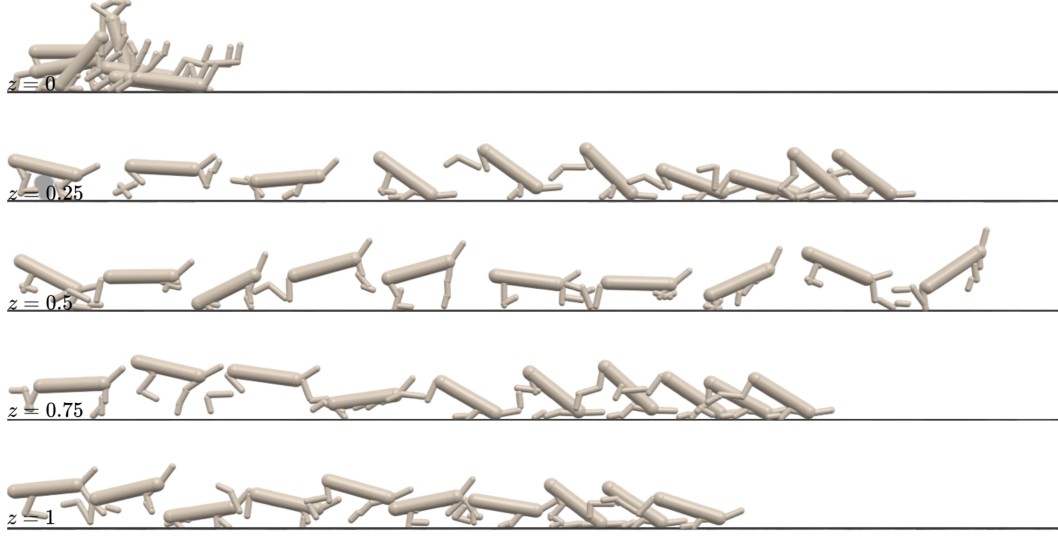

Figure 6: Extreme case: when torso radius and mass are increased by 50%. Only one policy is able to adapt without falling down ($z = 0.5$).

.

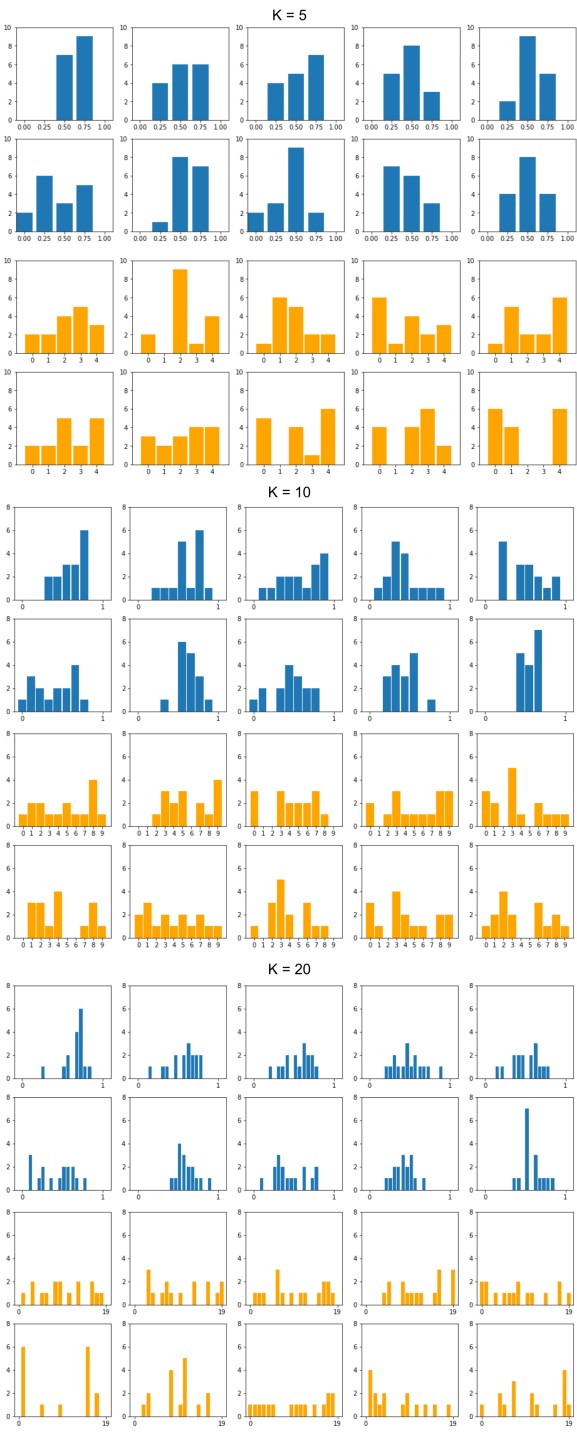

Figure 7: Number of times (y-axis) each policy (x-axis) is chosen by k-shot evaluation over the 16 test environments of the HalfCheetah for each of the 10 seeds (one table per seed). In blue, LoP, in orange, DIAYN+R. Please note that the 10 same LoP models are used for K=5, K=10, K=20 which is not the case for DIAYN+R.

.

Figure 8: We trained small discriminators over a dataset (100,000 environment interactions) of trajectories obtained with the learned policies of LoP and DIAYN+R when K=5. For each environment, for each seed, we trained a single discriminator and averaged the results. While the discriminators trained on DIAYN+R reach 100% accuracy rapidly on both train and test environments, they learn slower for LoP, with a slight advantage for the test environment, validating the fact that the diversity induced by the cosine similarity on the weights is more visible in variations of the environment rather than the environment on which the model has been trained. We evaluated the discriminator on a validation dataset (also 100,000 environment interactions) resulting in 100% accuracy for DIAYN in both train and test environments. For LoP, we obtained 82% accuracy on the training environment, and 87% on the test environments. The discriminator architecture consists in a neural network of two hidden layers of size 16, taking the unprocessed states as an input and outputting the predicted policy used (like in DIAYN).

.

Figure 9: Evolution of the best reward obtained with respect to K for LoP (N=2) and CoP (N=3) for each Halfcheetah test environment. We ran the K-shot evaluation for each K from K=1 to K=100 using the method described in Appendix A.2: we simply sample K random coefficients using the uniform distribution over $[0, 1]$ for LoP and the Dirichlet distribution over $[0, 1]^3$ for CoP. Results are averaged over 10 run for each K, and over the 10 models we learned for each method.

## B.2 ANT

Task originally coming from OpenAI Gym (Brockman et al. (2016)). Instead of using MuJoCo engine, we decided to use Brax (Freeman et al. (2021)) as it enables the possibility to acquire episodes on GPU. We use the vanilla environment for training. The policy and the critic are encoded by two different multi-layer perceptrons with ReLU activations. The base learning algorithm is PPO.

**Test environments:** As for HalfCheetah, we operated variations in physics (gravity and friction coefficients). We also designed environments with a percentage of masked features to simulate defective sensors (They are sampled randomly and remain the same for each run). Table 5 precisely indicates the nature of the changes for each environment

| Env name | Modifications |
|---|---|
| BigFriction | Friction coefficient $\times 1.25$ |
| BigGravity | Gravity coefficient $\times 1.25$ |
| SmallFriction | Friction coefficient $\times 0.75$ |
| SmallGravity | Gravity coefficient $\times 0.75$ |
| HugeFriction | Friction coefficient $\times 1.5$ |
| HugeGravity | Gravity coefficient $\times 1.5$ |
| TinyFriction | Friction coefficient $\times 0.5$ |
| TinyGravity | Gravity coefficient $\times 0.5$ |
| DefectiveSensor 5% | 5% of env obs set to 0 |
| DefectiveSensor 10% | 10% of env obs set to 0 |
| DefectiveSensor 15% | 15% of env obs set to 0 |
| DefectiveSensor 20% | 20% of env obs set to 0 |
| DefectiveSensor 25% | 25% of env obs set to 0 |
| DefectiveSensor 30% | 30% of env obs set to 0 |
| DefectiveSensor 35% | 35% of env obs set to 0 |

Table 5: Modified Ant environments used for testing.

| Hyper-parameter | Value |
|---|---|
| lr policy: | 0.0003 |
| lr critic: | 0.0003 |
| n parallel environments: | 2048 |
| n acquisition steps per epoch: | 20 |
| batch size: | 1024 |
| num minibatches: | 16 |
| update epochs: | 16 |
| discount factor: | 0.99 |
| clip ration: | 0.3 |
| action std: | 0.4 |
| gae coefficient: | 0.96 |
| reward scaling: | 1. |
| gradient clipping: | 10. |
| n layers (policy): | 4 |
| n neurons per layer (policy): | 64 |
| n layers (critic): | 5 |
| n neurons per layer (critic): | 256 |
| LoP | |
| $\beta$: | $\{0.1, 1, 10\}$ |
| DIAYN | |
| $\beta$: | $\{0.1, 1, 10\}$ |
| lr discriminator: | 0.001 |
| n layers (discriminator): | 2 |
| n neurons per layer (discriminator): | 64 |
| Lc | |
| $\beta$: | $\{0.1, 1, 10\}$ |
| dimensions of z: | 1 |
| lr discriminator: | 0.001 |
| n layers (discriminator): | 2 |
| n neurons per layer (discriminator): | 64 |

Table 6: Hyper-parameters for PPO over Ant

| | Single Policy | LoP | | | DIAYN + R | | | Lc | | |
|---|---|---|---|---|---|---|---|---|---|---|
| $\beta =$ | (K=1) | 0.1 | 1 | 10 | 0.1 | 1 | 10 | 0.1 | 1 | 10 |
| **K=5** | | | | | | | | | | |
| BigFriction | 7454 ± 166 | 7544 ± 140 | 7470 ± 96 | 7541 ± 202 | 7256 ± 1010 | 6403 ± 726 | 6267 ± 627 | **7666 ± 103** | 7573 ± 154 | 7538 ± 136 |
| BigGravity | 6905 ± 138 | 7038 ± 211 | 6937 ± 23 | 7027 ± 182 | 6858 ± 827 | 6082 ± 583 | 5865 ± 543 | **7123 ± 184** | 7075 ± 168 | 7051 ± 131 |
| SmallFriction | 6755 ± 2073 | 7695 ± 156 | 7652 ± 126 | 7599 ± 167 | 7046 ± 1777 | 6491 ± 807 | 6133 ± 706 | **7846 ± 127** | 7673 ± 183 | 7734 ± 211 |
| SmallGravity | 7057 ± 1875 | 7738 ± 134 | 7639 ± 120 | 7748 ± 169 | 7533 ± 896 | 6582 ± 727 | 6486 ± 769 | **7876 ± 100** | 7745 ± 105 | 7772 ± 84 |
| HugeFriction | 7505 ± 252 | 7634 ± 152 | 7616 ± 68 | 7601 ± 253 | 7220 ± 1331 | 6387 ± 752 | 6384 ± 733 | **7799 ± 103** | 7647 ± 128 | 7668 ± 77 |
| HugeGravity | 380 ± 507 | 1111 ± 538 | 1407 ± 557 | 1494 ± 934 | 846 ± 556 | **2924 ± 1992** | 2847 ± 1598 | 1134 ± 934 | 915 ± 465 | 1440 ± 985 |
| TinyFriction | 2747 ± 1241 | 3716 ± 996 | **4584 ± 801** | 4070 ± 1751 | 3540 ± 948 | 2950 ± 1113 | 2600 ± 572 | 3550 ± 843 | 4140 ± 294 | 3487 ± 508 |
| TinyGravity | -520 ± 426 | -106 ± 125 | -139 ± 274 | 116 ± 322 | -479 ± 374 | -3 ± 202 | **224 ± 108** | -234 ± 790 | -401 ± 373 | -83 ± 329 |
| DefectiveSensor 5% | 4308 ± 59 | 5243 ± 322 | **5630 ± 124** | 5560 ± 272 | 4958 ± 126 | 4492 ± 211 | 4360 ± 338 | 5428 ± 424 | 4988 ± 123 | 5225 ± 562 |
| DefectiveSensor 10% | 2770 ± 171 | 3620 ± 238 | **3660 ± 328** | 3625 ± 291 | 3440 ± 64 | 3371 ± 57 | 3223 ± 400 | 3519 ± 76 | 3406 ± 215 | 3491 ± 207 |
| DefectiveSensor 15% | 1531 ± 90 | 2583 ± 247 | 2738 ± 312 | **2740 ± 167** | 1774 ± 84 | 1984 ± 383 | 2055 ± 311 | 2408 ± 340 | 2226 ± 78 | 2349 ± 271 |
| DefectiveSensor 20% | 1026 ± 77 | 1750 ± 249 | 1965 ± 234 | **2008 ± 199** | 1104 ± 101 | 1490 ± 387 | 1450 ± 347 | 1658 ± 245 | 1546 ± 108 | 1714 ± 238 |
| DefectiveSensor 25% | 736 ± 95 | 1595 ± 317 | **1757 ± 154** | 1526 ± 338 | 762 ± 34 | 1107 ± 396 | 1120 ± 241 | 1344 ± 262 | 1271 ± 205 | 1363 ± 254 |
| DefectiveSensor 30% | 472 ± 50 | 695 ± 112 | 793 ± 166 | 674 ± 120 | 577 ± 34 | **859 ± 290** | 766 ± 212 | 673 ± 115 | 575 ± 49 | 622 ± 100 |
| DefectiveSensor 35% | 424 ± 48 | 606 ± 135 | **684 ± 182** | 635 ± 85 | 449 ± 68 | 650 ± 245 | 565 ± 151 | 618 ± 170 | 525 ± 92 | 602 ± 175 |
| **Average** | 3338 | 3905 | **4035** | 3998 | 3558 | 3451 | 3356 | 3909 | 3820 | 3870 |
| **K=10** | | | | | | | | | | |
| BigFriction | 7454 ± 166 | 7591 ± 134 | 7487 ± 101 | 7556 ± 199 | **7695 ± 113** | 6771 ± 573 | 6021 ± 748 | 7685 ± 103 | 7580 ± 163 | 7554 ± 116 |
| BigGravity | 6905 ± 138 | 7107 ± 153 | 7015 ± 89 | 7009 ± 166 | **7250 ± 57** | 6282 ± 476 | 5638 ± 600 | 7137 ± 96 | 7083 ± 184 | 7114 ± 98 |
| SmallFriction | 6755 ± 2073 | 7681 ± 154 | 7693 ± 103 | 7671 ± 193 | 7823 ± 142 | 6874 ± 590 | 5980 ± 716 | **7864 ± 131** | 7743 ± 161 | 7762 ± 196 |
| SmallGravity | 7057 ± 1875 | 7788 ± 135 | 7732 ± 86 | 7764 ± 185 | 7886 ± 127 | 6947 ± 619 | 6354 ± 818 | **7896 ± 91** | 7757 ± 117 | 7767 ± 75 |
| HugeFriction | 7505 ± 252 | 7640 ± 199 | 7589 ± 72 | 7613 ± 238 | **7860 ± 115** | 6788 ± 630 | 6160 ± 767 | 7846 ± 107 | 7695 ± 115 | 7668 ± 79 |
| HugeGravity | 380 ± 507 | 1329 ± 593 | 1711 ± 74 | 1583 ± 442 | 921 ± 302 | **4564 ± 2033** | 3009 ± 747 | 874 ± 358 | 1237 ± 662 | 1156 ± 812 |
| TinyFriction | 2747 ± 1241 | 4015 ± 1167 | **4918 ± 723** | 4393 ± 1252 | 4527 ± 560 | 3001 ± 1261 | 3676 ± 1382 | 4545 ± 731 | 4119 ± 555 | 3791 ± 476 |
| TinyGravity | -520 ± 426 | 55 ± 184 | 11 ± 226 | 154 ± 293 | -56 ± 256 | 6 ± 236 | **236 ± 336** | -149 ± 800 | -248 ± 362 | 227 ± 357 |
| DefectiveSensor 5% | 4308 ± 59 | 5088 ± 224 | 5323 ± 173 | 5098 ± 305 | 5063 ± 101 | 4498 ± 191 | 3992 ± 277 | **5355 ± 262** | 5016 ± 194 | 5229 ± 424 |
| DefectiveSensor 10% | 2770 ± 171 | 3974 ± 380 | **3978 ± 117** | 3934 ± 297 | 3486 ± 161 | 3413 ± 213 | 3061 ± 184 | 3960 ± 106 | 3711 ± 194 | 3779 ± 196 |
| DefectiveSensor 15% | 1531 ± 90 | **2504 ± 283** | 2423 ± 154 | 2496 ± 92 | 1851 ± 142 | 2365 ± 426 | 2226 ± 216 | 2309 ± 137 | 2374 ± 120 | 2352 ± 127 |
| DefectiveSensor 20% | 1026 ± 77 | 1795 ± 174 | 1629 ± 111 | 1774 ± 147 | 1205 ± 63 | 1777 ± 466 | 1737 ± 203 | 1834 ± 326 | **1987 ± 127** | 1787 ± 180 |
| DefectiveSensor 25% | 736 ± 95 | **1486 ± 156** | 1351 ± 220 | 1372 ± 85 | 851 ± 36 | 1422 ± 411 | 1380 ± 218 | 1342 ± 121 | 1368 ± 95 | 1390 ± 93 |
| DefectiveSensor 30% | 472 ± 50 | 1103 ± 87 | 968 ± 93 | 1133 ± 158 | 635 ± 99 | 967 ± 297 | 866 ± 157 | 1069 ± 144 | 925 ± 109 | 962 ± 149 |
| DefectiveSensor 35% | 424 ± 48 | **714 ± 149** | 640 ± 86 | 634 ± 75 | 441 ± 49 | **714 ± 223** | 662 ± 109 | 583 ± 95 | 609 ± 122 | 630 ± 107 |
| **Average** | 3338 | 3991 | **4031** | 4012 | 3833 | 3759 | 3400 | 4020 | 3947 | 3945 |
| **K=20** | | | | | | | | | | |
| BigFriction | 7454 ± 166 | 7592 ± 147 | 7510 ± 89 | 7569 ± 172 | 7646 ± 118 | 4328 ± 3086 | 5301 ± 429 | **7692 ± 94** | 7589 ± 129 | 7573 ± 122 |
| BigGravity | 6905 ± 138 | 7121 ± 162 | 7028 ± 50 | 7090 ± 171 | **7196 ± 139** | 4172 ± 2949 | 4867 ± 370 | 7162 ± 86 | 7135 ± 147 | 7132 ± 121 |
| SmallFriction | 6755 ± 2073 | 7767 ± 111 | 7712 ± 98 | 7703 ± 228 | 7833 ± 141 | 4360 ± 3118 | 5021 ± 322 | **7862 ± 131** | 7726 ± 144 | 7775 ± 170 |
| SmallGravity | 7057 ± 1875 | 7807 ± 123 | 7698 ± 86 | 7777 ± 203 | 7863 ± 109 | 4512 ± 3258 | 5343 ± 404 | **7906 ± 78** | 7774 ± 113 | 7806 ± 95 |
| HugeFriction | 7505 ± 252 | 7674 ± 171 | 7648 ± 97 | 7699 ± 198 | 7747 ± 104 | 4331 ± 3089 | 5061 ± 334 | **7851 ± 119** | 7714 ± 115 | 7737 ± 81 |
| HugeGravity | 380 ± 507 | 1655 ± 712 | 2111 ± 1405 | 1587 ± 343 | 2005 ± 743 | 3596 ± 2437 | **4231 ± 121** | 1357 ± 197 | 2122 ± 1006 | 1514 ± 845 |
| TinyFriction | 2747 ± 1241 | 4437 ± 827 | **5147 ± 595** | 4625 ± 1145 | 4393 ± 469 | 2203 ± 1432 | 3117 ± 816 | 4707 ± 490 | 3979 ± 656 | 3917 ± 529 |
| TinyGravity | -520 ± 426 | 259 ± 324 | 188 ± 241 | 289 ± 316 | **357 ± 525** | 291 ± 620 | 348 ± 152 | -121 ± 812 | -110 ± 257 | 177 ± 370 |
| DefectiveSensor 5% | 4308 ± 59 | 5816 ± 313 | 5996 ± 153 | 6002 ± 216 | 5185 ± 121 | 4570 ± 175 | 3647 ± 174 | **6023 ± 257** | 5780 ± 241 | 5884 ± 177 |
| DefectiveSensor 10% | 2770 ± 171 | 3979 ± 248 | 3850 ± 199 | 4083 ± 313 | 3500 ± 208 | 3475 ± 242 | 2827 ± 261 | **4173 ± 325** | 4023 ± 176 | 4038 ± 391 |
| DefectiveSensor 15% | 1531 ± 90 | 2789 ± 368 | 2526 ± 132 | 2790 ± 367 | 2047 ± 117 | 2495 ± 52 | 2125 ± 218 | 2596 ± 306 | **3011 ± 202** | 2810 ± 403 |
| DefectiveSensor 20% | 1026 ± 77 | 1890 ± 107 | 1794 ± 88 | 1750 ± 278 | 1333 ± 86 | 1875 ± 26 | 1714 ± 189 | 1746 ± 123 | 1789 ± 151 | **1901 ± 185** |
| DefectiveSensor 25% | 736 ± 95 | **1582 ± 104** | 1422 ± 187 | 1416 ± 150 | 992 ± 51 | 1351 ± 16 | 1320 ± 217 | 1480 ± 128 | 1286 ± 166 | 1450 ± 195 |
| DefectiveSensor 30% | 472 ± 50 | **1102 ± 133** | 1069 ± 128 | 968 ± 62 | 657 ± 75 | 931 ± 136 | 994 ± 70 | 909 ± 178 | 1035 ± 99 | 992 ± 51 |
| DefectiveSensor 35% | 424 ± 48 | **996 ± 103** | 908 ± 104 | 833 ± 77 | 485 ± 67 | 678 ± 15 | 719 ± 59 | 791 ± 100 | 927 ± 92 | 907 ± 113 |
| **Average** | 3338 | 4164 | **4174** | 4145 | 3949 | 2878 | 3109 | 4150 | 4126 | 4108 |

Table 7: Mean and standard deviation of cumulative reward achieved on Ant test sets per model. Results are averaged over 10 training seeds (i.e., 10 models are trained with the same hyper-parameters and evaluated on the 12 test sets). $K$ is the number of policies tested at adaptation time, using 1 episode per policy since this environment is deterministic. For this environment, we split the results per $\beta$ value as it has been used for beta ablation study (see Figure 3)

.

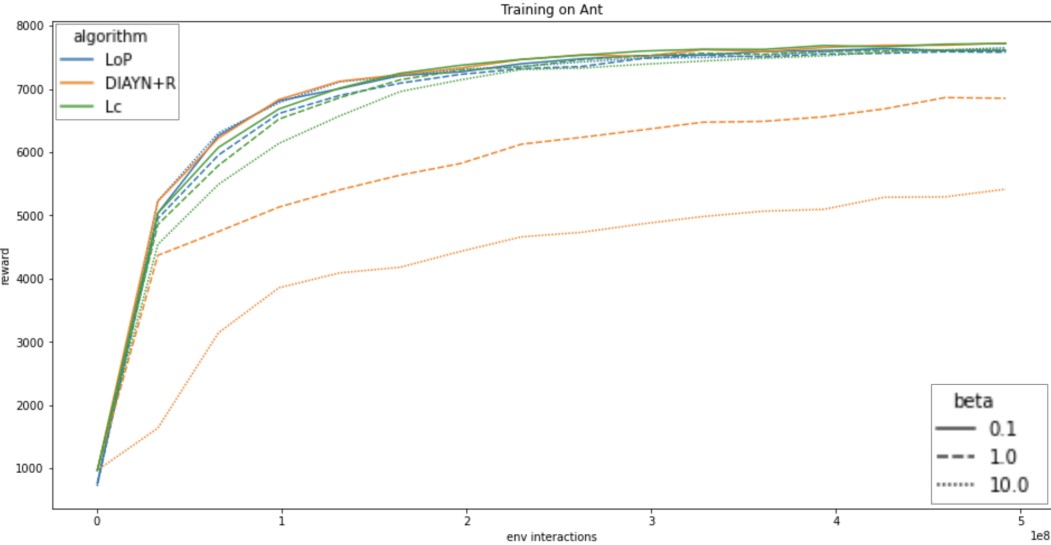

Figure 10: Evolution of the cumulative reward during training on the generic Ant environment for LoP, DIAYN+R and Lc for different values of beta. On can see that DIAYN+R struggles to perform well on the train set for $\beta = 1$ and $\beta = 10$. Results are averaged over 10 seeds.

## B.3 CARTPOLE

We use the openAI gym implementation of CartPole as a training environment. The 6 test environments are provided by Packer et al. (2018) where three different factors may vary: the mass of the cart, the length of the pole and the force applied to the cart. The length of each episode is 200. The policy and the critic are encoded by two different multi-layer perceptrons with ReLU activations. The base learning algorithm is A2C.

| Environment | Characteristics |
|---|---|
| (Train) CartPole | $mass = 0.1, length = 0.5, force = 10.0$ |
| HeavyPole CartPole | $mass = 1.0$ |
| LightPole CartPole | $mass = 0.001$ |
| LongPole CartPole | $length = 1.0$ |
| ShortPole CartPole | $length = 0.05$ |
| StrongPush CartPole | $force = 20.0$ |
| WeakPush CartPole | $force = 1.0$ |

Table 8: CartPole train and test environments

| Hyper-parameter | Value |
|---|---|
| learning rate: | 0.001 |
| n acquisition steps per epoch: | 8 |
| n parallel environments: | 32 |
| critic coefficient: | 1.0 |
| entropy coefficient: | 0.001 |
| discount factor: | 0.99 |
| gae coefficient: | 1.0 |
| gradient clipping: | 2.0 |
| n neurons per layer: | 8 |
| n layers: | 2 |
| LoP | |
| $\beta$: | 1.0 |
| DIAYN | |
| $\beta$: | 1.0 |
| n neurons per layer discriminator: | 8 |
| n layers discriminator: | 2 |
| learning rate discriminator: | 0.001 |

Table 9: Hyper-parameters for A2C over CartPole

| | Single | LoP | DIAYN+R | DIAYN+R $L_2$ |
|---|---|---|---|---|
| HeavyPole CartPole | 200.0 ± 0.0 | 200.0 ± 0.0 | 200.0 ± 0.0 | 200.0 ± 0.0 |
| LightPole CartPole | 200.0 ± 0.0 | 200.0 ± 0.0 | 200.0 ± 0.0 | 200.0 ± 0.0 |
| LongPole CartPole | 54.4 ± 81.6 | 56.1 ± 72.2 | 163.3 ± 73.3 | 123.8 ± 86.2 |
| ShortPole CartPole | 67.0 ± 33.1 | 78.9 ± 25.3 | 50.7 ± 18.1 | 64.8 ± 31.2 |
| StrongPush CartPole | 200.0 ± 0.0 | 200.0 ± 0.0 | 200.0 ± 0.0 | 199.9 ± 0.2 |
| WeakPush CartPole | 138.9 ± 43.8 | 164.4 ± 18.3 | 194.3 ± 10.0 | 148.1 ± 64.8 |
| Average | 143.4 | 149.9 | **168.1** | 156.1 |

Table 10: Results over CartPole, using 10 policies, and 10 episodes per policy at adaptation time.

## B.4 ACROBOT

We use the openAI gym implementation of Acrobot as a training environment. The 4 test environments are provided by Packer et al. (2018) where two different factor may vary: the intertia factor and the lengh of the system. We have used A2C as a learning algorithm.

| Environment | Characteristics |
|---|---|
| (Train) Acrobot | $mass = 0.1, length = 1.0, inertia = 1.0$ |
| Heavy Acrobot | $mass = 1.5$ |
| HighInertia Acrobot | $inertia = 1.5$ |
| Light Acrobot | $mass = 0.5$ |
| Long Acrobot | $length = 1.5$ |
| LowInertia Acrobot | $inertia = 0.5$ |
| Short Acrobot | $length = 0.5$ |

Table 11: Acrobot train and test environments

| Hyper-parameter | Value |
|---|---|
| learning rate: | 0.001 |
| n acquisition steps per epoch: | 8 |
| n parallel environments: | 32 |
| critic coefficient: | 1.0 |
| entropy coefficient: | 0.001 |
| discount factor: | 0.99 |
| gae coefficient: | 0.7 |
| gradient clipping: | 2.0 |
| n neurons per layer: | 16 |
| n layers: | 2 |
| LoP | |
| $\beta$: | 1.0 |
| DIAYN | |
| $\beta$: | 1.0 |
| n neurons per layer discriminator: | 16 |
| n layers discriminator: | 2 |
| learning rate discriminator: | 0.001 |

Table 12: Hyper-parameters for A2C over Acrobot

| | Single | LoP | DIAYN+R | DIAYN+R $L_2$ |
|---|---|---|---|---|
| Heavy Acrobot | -108.4 ± 3.2 | -105.1 ± 1.0 | -108.0 ± 1.8 | -108.2 ± 4.4 |
| HighInertia Acrobot | -108.7 ± 5.6 | -99.8 ± 2.8 | -106.0 ± 2.7 | -106.8 ± 8.9 |
| Light Acrobot | -120.7 ± 71.3 | -107.2 ± 58.8 | -115.2 ± 33.1 | -93.1 ± 37.3 |
| Long Acrobot | -124.3 ± 2.7 | -115.8 ± 9.6 | -117.3 ± 4.1 | -117.5 ± 5.3 |
| LowInertia Acrobot | -71.3 ± 2.3 | -70.7 ± 0.7 | -71.2 ± 0.7 | -71.3 ± 1.9 |
| Short Acrobot | -65.1 ± 2.8 | -60.7 ± 0.6 | -64.2 ± 2.6 | -64.7 ± 5.6 |
| Average | -99.7 | **-93.2** | -97.0 | -93.6 |

Table 13: Results over Acrobot, using 10 policies, and 10 episodes per policy at adaptation time.

We use the openAI gym implementation of Pendulum as a training environment. The 3 test environments are provided by Packer et al. (2018) where two different factor may vary: the mass and the length of the pendulum. We have considered 5 discrete actions between $-1$ and $+1$. We have used A2C as a learning algorithm.

| Environment | Characteristics |
|---|---|
| (Train) Pendulum | $mass = 1.0, length = 1.0$ |
| Light Pendulum | $mass = 0.5$ |
| Long Pendulum | $length = 1.5$ |
| Short Pendulum | $length = 0.5$ |

Table 14: Pendulum train and test environments

| Hyper-parameter | Value |
|---|---|
| learning rate: | 0.001 |
| n acquisition steps per epoch: | 8 |
| n parallel environments: | 32 |
| critic coefficient: | 1.0 |
| entropy coefficient: | 0.001 |
| discount factor: | 0.99 |
| gae coefficient: | 0.7 |
| gradient clipping: | 2.0 |
| n neurons per layer: | 16 |
| n layers: | 2 |
| LoP | |
| $\beta$: | 1.0 |
| DIAYN | |
| $\beta$: | 1.0 |
| n neurons per layer discriminator: | 16 |
| n layers discriminator: | 2 |
| learning rate discriminator: | 0.001 |

Table 15: Hyper-parameters for A2C over Pendulum

| | Single | LoP | DIAYN+R | DIAYN+R $L_2$ |
|---|---|---|---|---|
| Light Pendulum | -36.5 ± 58.5 | -11.4 ± 2.7 | -39.3 ± 10.9 | -32.1 ± 15.4 |
| Long Pendulum | -82.1 ± 20.4 | -64.5 ± 13.0 | -70.6 ± 15.2 | -71.9 ± 17.3 |
| Short Pendulum | -39.6 ± 66.9 | -10.7 ± 2.1 | -31.3 ± 11.7 | -28.2 ± 13.3 |
| Average | -52.7 | **-28.9** | -47.1 | -44.0 |

Table 16: Results over Pendulum, using 10 policies, and 10 episodes per policy at adaptation time.

## B.6 MINIGRID

We have use Gym Minigrid to perform experiments on mazes Chevalier-Boisvert et al. (2018). We have used the MultiRoom-N2-S4 for training considering one single maze. At test time, we have tested on three different MultiRoom-N2-S4 environments composed of two rooms, but also on three MultiRoom-N4-S5 composed of four rooms. This allows us to evaluate the generalization power of the different methods to larger mazes. We have used A2C as a learning algorithm.

| Hyper-parameter | Value |
|---|---|
| learning rate: | 0.001 |
| n acquisition steps per epoch: | 8 |
| n parallel environments: | 32 |
| critic coefficient: | 1.0 |
| entropy coefficient: | 0.001 |
| discount factor: | 0.99 |
| gae coefficient: | 0.7 |
| gradient clipping: | 2.0 |
| n neurons per layer: | 16 |
| n layers: | 2 |
| LoP | |
| $\beta$: | 1.0 |
| DIAYN | |
| $\beta$: | 1.0 |
| n neurons per layer discriminator: | 16 |
| n layers discriminator: | 2 |
| learning rate discriminator: | 0.001 |

Table 17: Hyper-parameters for A2C over Minigrid

| | Single | LoP | DIAYN+R | DIAYN+R $L_2$ |
|---|---|---|---|---|
| Two Rooms Maze 1 | 0.387 ± 0.447 | 0.619 ± 0.309 | 0.348 ± 0.348 | 0.656 ± 0.328 |
| Two Rooms Maze 2 | 0.433 ± 0.499 | 0.865 ± 0.0 | 0.627 ± 0.363 | 0.692 ± 0.346 |
| Two Rooms Maze 3 | 0.194 ± 0.387 | 0.617 ± 0.309 | 0.348 ± 0.353 | 0.656 ± 0.328 |
| Four Rooms Maze 1 | 0.0 ± 0.0 | 0.294 ± 0.36 | 0.0 ± 0.0 | 0.24 ± 0.359 |
| Four Rooms Maze 2 | 0.0 ± 0.0 | 0.004 ± 0.007 | 0.0 ± 0.0 | 0.137 ± 0.274 |
| Four Rooms Maze 3 | 0.0 ± 0.0 | 0.281 ± 0.345 | 0.166 ± 0.287 | 0.28 ± 0.345 |
| Average | 0.169 | **0.447** | 0.248 | 0.443 |

Table 18: Results over Minigrid, using 10 policies, and 1 episode per policy at adaptation time.

## B.7 PROCGEN (FRUITBOT)

We performed experiments on pixel-based environment with ProcGen. We used the FruitBot game for training considering 10 levels sampled uniformly at each episode. At test time, we selected 3 different environments, each of them composed of 10 uniformly sampled levels, sampled uniformly, and that were not seen at train time. We used the CNN architecture described in the ProcGen paper (IMPALA architecture (Espeholt et al. (2018))). For LoP, the two first blocks are fixed and only the parameters of the last block depend on the value of $z$. For DIAYN, the $z$ value is provided as a one-hot vector stacked to the observation. As for Brax environment, we used PPO algorithm.

| Hyper-parameter | Value |
|---|:---:|
| learning rate: | $5e-4$ |
| n acquisition steps per rollout: | 128 |
| n batches epoch: | 8 |
| n epochs per rollout: | 5 |
| n parallel environments: | 64 |
| critic coefficient: | 1.0 |
| entropy coefficient: | 0.01 |
| discount factor: | 0.99 |
| gae coefficient: | 0.0 |
| gradient clipping: | 20.0 |
| LoP | |
| $\beta$: | $0.01, 0.1, 1.0$ |
| DIAYN+R | |
| $\beta$: | $0.001, 0.01, 0.1, 1.0$ |

Table 19: Hyper-parameters for PPO over ProcGen

| | Single | LoP | DIAYN+R |
|---|:---:|:---:|:---:|
| Levels 100 to 110 | 11.9 ± 2.6 | 20.1 ± 4.4 | 12.1 ± 5.2 |
| Levels 200 to 210 | 7.1 ± 1.7 | 10.3 ± 0.2 | 5.1 ± 1.7 |
| Levels 300 to 310 | 14.3 ± 5.1 | 18.7 ± 2.7 | 17.1 ± 4.6 |

Table 20: Results over Procgen, using K=10 at adaptation time, (averaged over 16 episodes per shot), averaged over 5 runs.

## B.8 MAZE 2D WITH WALLS

To visualize the policies learned by the different methods, we just implemented a simple discrete maze (4 actions = up,down, left, right) where the objective is to go from the top-middle tile to the bottom-middle tile by moving through a corridor (size is $21 \times 11$). Reward is -1 at each step until goal is reached and the maximum number of steps is $-100$. The optimal policy in the training environment achieves $-16$. At test time, we generate walls in the corridor such that the agent has to avoid these walls to reach the goal. The observation space is a $5 \times 5$ square around the agent. Policies are learned by using PPO. Illustrations of the trajectories over the train and the 4 test environments are illustrated in Figures 11,12,13,14

| Hyper-parameter | Value |
|---|---|
| learning rate: | 0.001 |
| n acquisition steps per rollout: | 16 |
| n batches epoch: | 4 |
| n epochs per rollout: | 3 |
| n parallel environments: | 32 |
| critic coefficient: | 1.0 |
| entropy coefficient: | 0.01 |
| discount factor: | 0.99 |
| gae coefficient: | 0.0 |
| gradient clipping: | 20.0 |
| **LoP** | |
| $\beta$: | $0.01, 0.1, 1.0$ |
| **DIAYN+R** | |
| $\beta$: | $0.01, 0.1, 1.0$ |

Table 21: Hyper-parameters for PPO used for Maze 2d

| | Single Policy | LoP | | | DIAYN + R | | |
|---|---|---|---|---|---|---|---|
| $\beta =$ | | 0.01 | 0.1 | 1. | 0.01 | 0.1 | 1. |
| Test Env #1 | -16.0 ± 0.0 | -17.6 ± 2.2 | -17.2 ± 1.1 | -17.6 ± 2.6 | **-16 ± 0** | -16 ± 0 | -16 ± 0 |
| Test Env #2 | -100 ± 0.0 | **-23.6 ± 3.6** | -39.4 ± 25.8 | -30.2 ± 11.6 | -56.2 ± 40.2 | -42.6 ± 33 | -44.8 ± 31.5 |
| Test Env #3 | -100 ± 0.0 | **-37.8 ± 18.9** | -41.6 ± 10.6 | -43.6 ± 11.5 | -54.2 ± 27.1 | -55.8 ± 31.4 | -60.4 ± 27 |
| Test Env #4 | -100 ± 0.0 | **-42.2 ± 11.8** | -60.8 ± 27.9 | -48 ± 22.9 | -54.8 ± 28.9 | -53.4 ± 26.5 | -72.8 ± 33.8 |
| Test Env #4 | -100 ± 0.0 | **-28.4 ± 10.7** | -44.4 ± 30.2 | -28.8 ± 8.9 | -37.4 ± 35.4 | -43.2 ± 34.7 | -50.4 ± 38.8 |
| **Average** | -83.2 | **-29.9** | -40.68 | -33.64 | -43.72 | -42.2 | -48.88 |

Table 22: Results over Maze2d, using K=10, averaged over 5 runs.

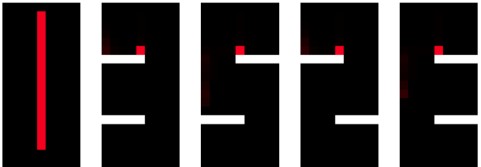

Figure 11: Trajectories learned by a Single Policy. First column is the training environment. Other columns are test environments. The lighter red the tiles are, the longer the agent stays on a particular location.

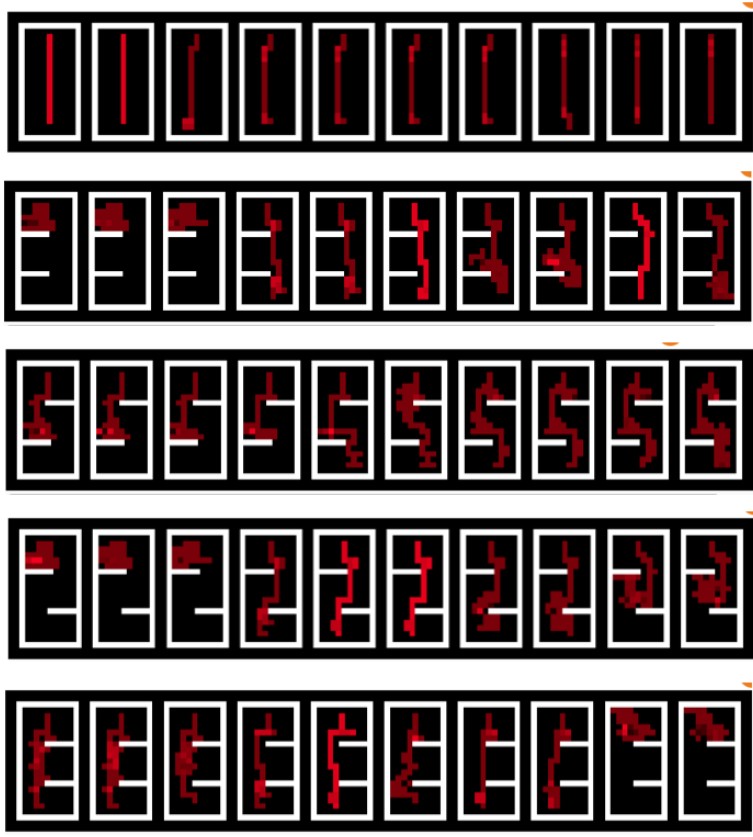

Figure 12: Trajectories learned by LoP ($\beta = 1.0$). Rows are environments, columns are the $K = 10$ policies test during online adaptation. For each test environment, at least one policy is able to reach the goal

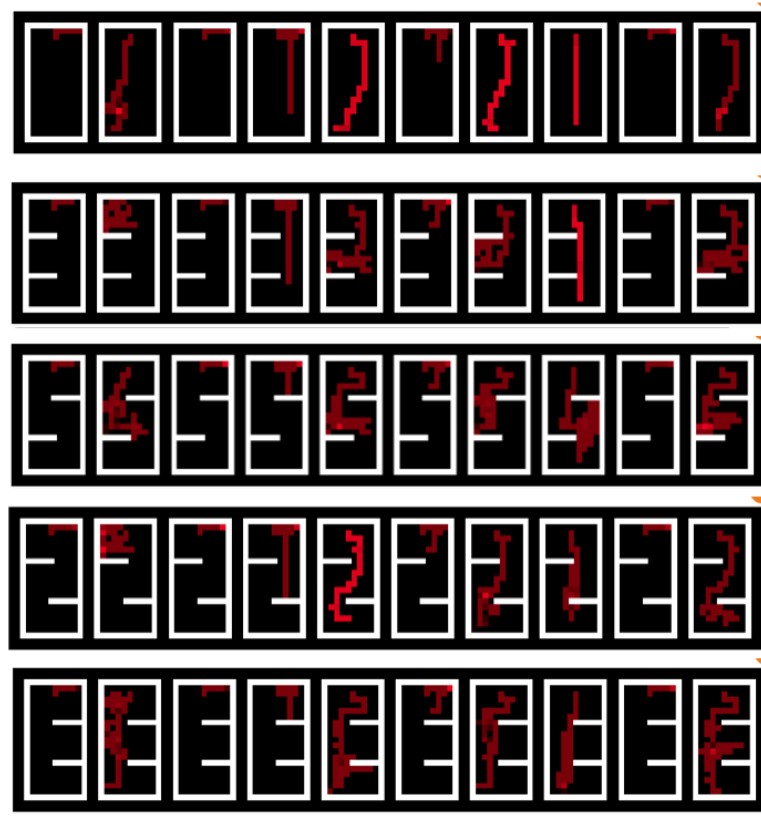

Figure 13: Trajectories learned by DIAYN+R ($\beta = 0.01$). Rows are environments, columns are the $K = 10$ policies test during online adaptation. For many test environment, at least one policy is able to reach the goal

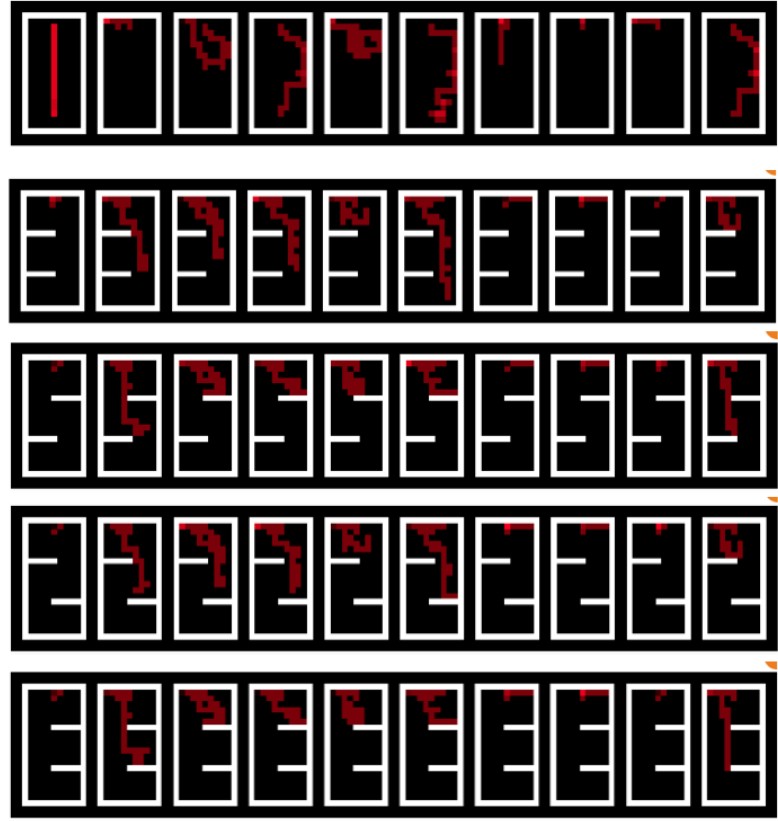

Figure 14: Trajectories learned by DIAYN+R ($\beta = 1$). Rows are environments, columns are the $K = 10$ policies test during online adaptation. With a too high value of $\beta$, the training policy is suboptimal, and does not achieve good performance at train and test times.

