# OpenReview forum: "Learning a subspace of policies for online adaptation in Reinforcement Learning"
_ICLR.cc/2022/Conference — ICLR 2022 Poster_

### Official Review · Reviewer_qrCv · 2021-10-27

**Correctness:** 4
**Technical Novelty And Significance:** 2
**Empirical Novelty And Significance:** 3
**Recommendation:** 8
**Confidence:** 5

**Main Review:**

I enjoyed reading this paper, it studies an important problem and proposes new methods to tackle it. Thats said, I have some questions about the proposed new ideas in this paper and the connection to related work. I also have a few questions about the experiments.

Regarding the line of policies. Let's assume for a moment that the anchor policies are deterministic, and consider the state occupancies of these policies. Using standard arguments (see [1] for example), we will get that any policy along the line (or the convex Hull) is a stochastic (or mixed) policy, and in addition, that for any reward signal, one of the deterministic policies will always be at least as good than any of the stochastic policies. That is, the anchor policies will be at least as good as any policy along the line. Now, there are a few differences between this construction and the approach taken in the paper, in particular, since the line is in the parameter space and not in the policy/state occupancy space. Nevertheless, I would like to see some evidence that the non-anchor policies are actually used over the anchor policies and to understand why that is. I am also very interested to see if the line in the parameter space extrapolates the convex Hull in the state occupancy space. This should be correlated with having policies along the line that outperform the anchor policies for some reward signals.

Regarding the experiments, the results are very impressive. One thing that I am missing is the data about performance during training time, and in particular, what is the average reward and diversity score (e.g. the cosine similarity or mutual information) achieved. This is important, since a potential scenario in Cheetah might be that a set that is more diverse but less good in terms of reward is leading to better fast adaptation. If this is the case (if one has to sacrifice reward to quickly adapt better) then I would like to learn that.

One baseline that I am missing, and follows the same line of thinking is a constrained MDP approach as was taken in [2,3]. The idea here is to set a constraint on how much a method will satisfy the extrinsic reward and then to maximize diversity under this constraint. Also note that there is a strong connection between the cosine similarity and the average reward in [3], but in [3] it is an inner product (not dividing by the norm) and being used in the successor features space instead of the parameter space.

[1] Zahavy, T., Barreto, A., Mankowitz, D.J., Hou, S., O'Donoghue, B., Kemaev, I. and Singh, S., 2020, September. Discovering a set of policies for the worst case reward. In International Conference on Learning Representations.

[2] Kumar, S., Kumar, A., Levine, S. and Finn, C., 2020. One Solution is Not All You Need: Few-Shot Extrapolation via Structured MaxEnt RL. Advances in Neural Information Processing Systems, 33.

[3] Zahavy, T., O'Donoghue, B., Barreto, A., Flennerhag, S., Mnih, V. and Singh, S., 2021, June. Discovering diverse nearly optimal policies with successor features. In ICML 2021 Workshop on Unsupervised Reinforcement Learning.

**Summary Of The Paper:**

The paper studies how to discover a subset of policies that will become useful for fast adaptation of future tasks. The subspace is defined as the convex hull of some anchor policies which are discovered, and are encouraged to perform well with respect to the extrinsic reward while being as different as possible from each other in the parameter space. The quality of the set is then evaluated by its ability to quickly adapt to new tasks.

**Summary Of The Review:**

Nice paper, some questions about the proposed idea needs answering and some of the experimental results need further clarification.

---

> ### Author Response · Authors · 2021-11-19
> **Answer from authors**
>
> Thank you for your remarks. Please have a look at the “answer to all reviewers”.
>
> * **Evidences that the non-anchor policies are actually used:** We have added an analysis of which policies are used depending on the test environments (see Figure 4 (right) and Figure 7 in appendix B1). This analysis shows that a) anchor policies are little used in the test environments and b) different test environments make use of different policies (i.e., different z). We hypothesize that it is a border effect of the training objective we defined : as we aim to optimize the whole line segment at train time (see Algorithm in Figure 2), by randomly sampling a z value at each epoch, the middle of the line segment should benefit from both anchor policies updates while these anchor policies benefit from the update of themselves. This is something that has also been analyzed by [Wortsman et al. 2021] in supervised learning, with clearer evidence that the midpoint always outperforms the other points in the line segment.
>
>
> * **Performance during training time** is illustrated in Figure 3 (left) and in Figure 10 in appendix B2 for the training curves. It shows that our method is as sample efficient as the DIAYN-based methods. We have performed new experiments on the Procgen environments where the policy architecture is larger than in MUJOCO (pixel-based RL). In these environments, LoP needs more training iteration (about twice) than a single policy to converge which makes sense, but less iterations than a DIAYN approach. You can see the results and implementation details in appendix B7.
>
>
> * **Concerning additional baselines**, we did not perform comparisons with SMERL as explained in Section 4 for the exact same reasons as what is stated in [Osa et al. 2021]: SMERL actually needs to be aware of the complete episode rewards which is not obvious in terms of implementation with PPO. We were not aware of the paper [3] which is very recent (thanks for the ref), and we don’t have time to compare with the proposed model during the rebuttal period since there is no available open-source implementation.  We expect to have comparisons with this paper in the final version if accepted.
>
>
> Thanks for your remarks, and happy to discuss more if you have other remarks

---

> > ### Comment · Reviewer_qrCv · 2021-11-26
> > **Follow up**
> >
> > Thank you for the detailed reply and for running those additional experiments, I think they help me to understand Lop in more detail.
> > I have two follow up questions on the first bullet after looking at figures 4 and 7. Firstly, I can see that the y axis corresponds to the number of times a policy is being chosen. I am also interested in the reward associated with these policies, is it possible to add this statistic? the reason for that is that its possible that one policy is being selected while other policies achieve very similar reward. My second question is about Figure 7, as you increase the number of policies, the plot for which policy is being selected starts to look more and more like a Gaussian, centred around the middle of the line. Any idea why the middle is preferred over the anchors or over other points across the line?
> >
> > Regarding my second bullet, can you elaborate more on the evaluation procedure? in particular, the reward of which policy is being reported here during training? is it the anchor policies? or the random policies that are being sampled from the line during training?

---

> > > ### Author Response · Authors · 2021-11-26
> > > **Follow up**
> > >
> > > Thank you for your additional comments. About your follow up questions:
> > >
> > > **Reward associated with the policies at test time:**  We displayed several trajectories with their associated reward of the same LoP model for several extreme environments on our website:
> > >
> > > - [Huge Torso](https://sites.google.com/view/subspace-of-policies/home/huge-torso)
> > > - [Disproportionate feet](https://sites.google.com/view/subspace-of-policies/home/disproportionate-feet)
> > > - [Extreme gravity](https://sites.google.com/view/subspace-of-policies/home/halfcheetah-gravity)
> > > - [Extreme friction](https://sites.google.com/view/subspace-of-policies/home/extreme-friction)
> > > - [Thin legs](https://sites.google.com/view/subspace-of-policies/home/thin-legs)
> > >
> > > The discrepancy in terms of rewards is high in these examples. In addition, we added the [detailed results](https://sites.google.com/view/subspace-of-policies/home/results) of this LoP model, including the training environment (we added the table below). One can see that the more extreme the environment variations are, the higher the discrepancy is.
> > >
> > > |                   	|      0 	|   0.25 	|    0.5 	|   0.75 	|      1 	|
> > > |-------------------	|------:	|------:	|------:	|------:	|------:	|
> > > | Classical (train) 	| 10712 	| 10696 	| 11033 	| **11627** 	| 10826 	|
> > > | BigFeet           	|  7712 	|  8554 	|  1781 	|  **8717** 	|  7480 	|
> > > | BigFriction       	| 10920 	| 11230 	| **11436** 	| 11257 	|  1109 	|
> > > | BigGravity        	|  **9899** 	|  9848 	|  9379 	|  9637 	|  4643 	|
> > > | BigShins          	|   371 	|  **8757** 	|  8550 	|  8512 	|  7817 	|
> > > | BigThigs          	|  9427 	|  **9965** 	|  9257 	|  9253 	|  8452 	|
> > > | BigTorso          	|   807 	|  **9862** 	|  9120 	|  8930 	|  8918 	|
> > > | HugeFriction      	| 10260 	| **11070** 	|  3167 	|  1287 	|  1118 	|
> > > | HugeGravity       	|  **8329** 	|  8242 	|  2393 	|  -126 	|  6732 	|
> > > | SmallFeet         	|  7864 	|  **7895** 	|  6805 	|  7135 	|  7036 	|
> > > | SmallFriction     	|  1238 	| 10575 	| 10865 	| **10937** 	| 10076 	|
> > > | SmallGravity      	| 10405 	| 11081 	| **11830** 	| 11773 	| 11040 	|
> > > | SmallShins        	|  9984 	| **11093** 	| 10484 	|  9796 	|  8892 	|
> > > | SmallThigs        	| 10183 	| 10411 	| 11298 	| **11352** 	|  9495 	|
> > > | SmallTorso        	| 10552 	| 10913 	| **11240** 	| 11220 	|  9796 	|
> > > | TinyFriction      	|  9723 	|  9969 	|  9914 	| **10232** 	|  8976 	|
> > > | TinyGravity       	|  9584 	|  3883 	| 11164 	| **11353** 	|  3050 	|
> > >
> > > **Why is the middle preferred over the anchors or over other points across the line?**  It has also been pointed out by [Wortsman et al. 2021] in supervised learning, without further explanation though. One possible explanation is that - if we see mode connectivity as an implicit ensemble method - the points in the middle of the line segment would be the ones that benefit the most from the updates of both $\bar{\theta}_1$ and $\bar{\theta}_2$, while the extremities are more “attracted” by the update of the nearest anchor policy, and thus tend to overfit like a single policy would do.
> > >
> > > **Can you elaborate more on the evaluation procedure?**  The reward reported during training in Figure 3 (left - right column) is the average reward of 256 (deterministic) policies uniformly sampled over the policy line on the training environment (test environments are not available at train time). The training curves are available in Figure 10 (Appendix B.2). It is important to note that the average performance obtained by LoP (at train time) is very similar to the performance obtained by a single policy trained over the training environment (paper will be updated to make this result appears)
> > >
> > >
> > > Thanks for your remarks, and happy to discuss more if you have other remarks.

---

> > > > ### Comment · Reviewer_qrCv · 2021-11-29
> > > > **Response**
> > > >
> > > > I thank the authors for answering all my questions and providing additional information during the rebuttal period. When I first read the paper I felt that the LOP is a strong idea and novel in the RL setting. That said, while it is the main contribution of the paper I also felt that it was under developed/analysed. However, the additional data provided by the authors as well as the discussion and explanations around it during the rebuttal improve the understanding and analysis of the LOP significantly. For me, this is enough to recommend an accept and I will increase my score accordingly.

---

> > > > > ### Comment · Reviewer_qrCv · 2021-11-29
> > > > > **Follow up**
> > > > >
> > > > > I do have one more follow up question after taking a look at the Wortsman et al. 2021 paper. There, the authors indeed observed that the middle point consistently outperform the anchor points when no assembling is used, however, they also observed that when an ensemble is used, the best performance is achieved closer to the anchors, and that the overall performance is better with an used rather than without it. However, as far as I understand, no ensemble has been used in the current paper.
> > > > >
> > > > > Can you elaborate more on that? was this something that you tried but didn't work? is there a significant difference between the RL/fast adaptation setup to the SL setup in Wortsman et al. 2021 that explains that?

---

> > > > > > ### Author Response · Authors · 2021-11-29
> > > > > > **Follow up**
> > > > > >
> > > > > > Hi,
> > > > > >
> > > > > > In all the MUJOCO/BRAX environments, actions are continuous. Since we are using PPO, we obtain a policy that is (at test-time) not stochastic. In that case, it is not obvious to do ensembling (ensembling in the supervised classification setting is about averaging the output distribution over classes) and this is the main reason why we do not propose results using ensembling. Indeed, we are not sure that averaging directly the actions make sense, and averaging over continuous action distribution would need to move for instance from PPO to SAC where explicit distribution are learned. For environments with discrete actions, such results could be included, but we did not make experiments in that direction. One other aspect is that, for online adaptation, ensembling would consist in finding the best ensemble of policies, thus increasing the search complexity (we would have to evaluate episodes with multiple ensembles of policies) and slowing down the adaptation. In general, we think that there is a high relationship between ensembling and subspaces of models/policies that we plan to explore in the next months.

---

> ### Author Response · Authors · 2021-11-25
> **answer**
>
> Hello, we would be grateful if you can confirm whether our response has addressed your comments, and let us know if any issues remain.

---

### Official Review · Reviewer_D4JT · 2021-11-02

**Correctness:** 3
**Technical Novelty And Significance:** 3
**Empirical Novelty And Significance:** 4
**Recommendation:** 6
**Confidence:** 4

**Main Review:**

The paper is generally well-written and easy to follow.

The proposed method is motivated by a robustness argument which is intuitive but could benefit from further exploration. While it is easy to believe that parameters that are robust to perturbations will perform well when the model is perturbed, this is not obviously true (in fact, one can easily construct examples where this does not hold). It is also unclear why this property would be preserved if the robustness is limited to a 1-dimensional line in parameter space. Additional explanations would be especially welcome as the baselines used in this paper are based on diversity and thus follow a different intuition. Furthermore, the authors should situate this work within the broader literature of robust reinforcement learning which has existed for decades.

The evaluation on multiple variations of control domains is promising and shows that the method can lead to improved adaptation with a relatively simple training procedure. The analysis on hyper-parameter sensitivity is welcome and it is good to see that the method does not require a lot of fine-tuning. Overall, the analysis focuses on relatively easy problems for adaptation and it would be interesting to see how this scales to harder tasks where a non-adapted policy fails completely and to analyze at which point the method stops being able to adapt. It would also be interesting to see a characterization of the learned sub-space, i.e. what kind of policies are being learned by the agent and how do they qualitatively compare to the kind of policies learned by diversity objectives. Finally, while increasing K leads to apparent improvements, it would also be interesting to see how the method compares to regular reinforcement learning when K=0, i.e. if we see a benefit from finding robust parameters, even if we do not optimize within the learned subspace.

In the paragraph on K-shot adaptation (Section 3.1), the authors claim that only one value of z needs to be executed in a deterministic environment. This is only true if the learned policy is also deterministic which does not seem to be the case in the proposed set-up.

In Section 3.2, the authors claim that the proposed method does not lead to learning sub-optimal policies at train-time. This statement is too strong as there is still a trade-off. In particular, the objective encourages the agent to find wide local optima and may forgo a better, but more narrow local optimum that it would otherwise choose.

In Section 2 on notation, r(s,a) should map to R, not R+.


**Summary Of The Paper:**

The authors propose a method for rapid adaptation from a single training task to an unseen test-task in reinforcement learning. The method optimizes to find a convex subspace, specifically a line, of parameters that minimize the objective in expectation over a uniform distribution. For adaptation, the authors propose to find a convex combination that performs well on the test task.

**Summary Of The Review:**

The paper is generally well-written and easy to follow. The problem formulation proposed by the authors is both relevant and important and the proposed method is simple to implement while showing promising results; however, the paper could benefit from additional analysis and more comprehensive literature review covering robust reinforcement learning.

---

> ### Author Response · Authors · 2021-11-19
> **Answer from authors**
>
> Thank you for your remarks. Please have a look at the “answer to all reviewers”.
>
> * **Explanation regarding the robustness of the method and its positioning regarding robust RL:** We agree that we did not position our paper w.r.t robust RL literature. This has been fixed in the new version of the paper (Section 6, last paragraph).  The main difference is that Robust RL aims at learning policies efficient ‘in the worst case’ and is usually not about adaptation to test environments (the objective is to learn a single policy which is robust, while we are learning multiple policies, and then selecting the one that is the best at test time)
>
>
> * **Experiments on harder tasks:** In Table 7 in Appendix, we have explored the ability of our method to handle different amounts of corrupted input features. These experiments clearly show that, at some point, when the variation is ‘too big’ our model cannot adapt to the testing environment (like other baselines). In addition, we have performed new experiments on the ProcGen environment to see if our model is able to handle large architectures used for pixel-based RL (Table 1 and Appendix B.7). Results show that LoP behaves also well in this environment.  This confirms the effectiveness of our approach to different types of variations, including noisy features on which baselines methods were not applied in previous publications. Note that our experimental setting is limited by the amount of computations (see  “answer to all reviewers”).
>
>
> * **Value of z for deterministic environment:** At test time, we used the deterministic version of the learned policies (i.e. picking the argmax of the output at each timestep). We clarify this point in the new version of the paper.
>
>
> * **Section 3.2, strong assumption about learning sub-optimal policies:** We agree that, since we are using an auxiliary loss, it may prevent the training algorithm from discovering good training policies. But, in our experiments, we have observed that, even when the auxiliary loss reaches 0 (i.e the diversity in the parameters space constraint is fulfilled), the training performance was similar to the one obtained by a single policy trained without constraints. In the DIAYN+R method, satisfying the diversity constraint is detrimental to the train performance (see the training curves in Figure 8 in Appendix B2). Said otherwise, our constraint is easier to satisfy than the functional diversity constraint, making our model much more easier to tune. We have reformulated this claim in the article to make it more clear.
>
>
> * **Notations:** Thanks for pointing this out. This has been corrected.
>
> Thanks for your remarks, and happy to discuss more if you have other remarks

---

> ### Author Response · Authors · 2021-11-25
> **asnwer**
>
> Hello, we would be grateful if you can confirm whether our response has addressed your comments, and let us know if any issues remain.

---

### Official Review · Reviewer_UNwx · 2021-11-03

**Correctness:** 3
**Technical Novelty And Significance:** 3
**Empirical Novelty And Significance:** 3
**Recommendation:** 6
**Confidence:** 4

**Main Review:**


The problem is interesting, and the proposed method is simple and apparently easy to tune. However, I am concerned about whether or not this method will scale to more challenging environments than the Gym benchmark tasks because of various qualities of the method I describe below. While this is a promising direction of research, I think that this paper requires some iteration before acceptance.

Strengths:

- The proposed method is very simple, and seems to be relatively simple to optimize hyperparameters
- The empirical results seem to convincingly indicate that this approach outperforms prior work
- I appreciated Figure 1a: it neatly conveys the main approach of the paper

Weaknesses:

- While the authors seem to try to argue otherwise, it appears that enabling generalization to new test environments requires diversity in behavior / state distribution ($\pi(a|s)$ or $d^\pi(s)$) and not diversity in parameter space (since if parameters differ, but result in the same function, then there is no benefit conferred). However, unlike prior work (e.g. Kumar et al 2020b), this method does not optimize for functional diversity, and only parameter diversity, and in fact there is nothing in the loss that prevents the objective learning the same policy (e.g. in a "flat" part of the parameter space). No analysis is done as to how functionally diverse the learned policies actually are in the test environments as well (it would be interesting to see that)
- It seems concerning that the optimal instantiation of this method is with $n=2$ anchor points, effectively only leading to a "line" of policies, and not the more general subspace of policies alluded to in the introduction / title. While there is a little discussion with n=3 in the experiments section, I think the paper would be stronger if it analyzed more deeply why this approach does not do well with more anchors. As tasks get more complicated, it seems that the "line" of policies with only one factor of variation will be unlikely to capture the variation needed to perform well in a shifted test environment.
- An important missing comparison is just learning an ensemble of policies (either w/ or w/out standard diversity based metrics), and using the best ensemble member in the test environment. While this (hopefully) should not outperform the proposed method, this comparison would be useful to understand how much the continuous space of policies helps the algorithm, over simply just having a diverse set of policies.
- In the experiments section, it says that the hyperparameters for the comparison methods are tuned based on performance in the training environment. It was unclear to me if training environment performance was used for tuning $\beta$ for the DIAYN comparisons, since later it also says that the results provided are for the best value of $\beta$ (which seems to imply test env performance was used). I would appreciate this if this were clarified.
- Is there a way to tune the method depending on how different we expect the test-time environment to be? Abstractly speaking, if we know that $D(\mathcal{M}, \hat{\mathcal{M}}) < \epsilon$, how should we choose the hyperparameters for this algorithm? It seems reasonable to expect our method to be more conservative when the test environment is expected to be far from the current one, and less conservative when not. My original reaction was that $\beta$ in the equation after Eq5 may do this, but it seems in your experiments that the constraint $C(\cdot, \dots, \cdot)$ goes to zero regardless, so the behaviour would not change for other values of $\beta$? Please let me know if I am misunderstanding.
- The particular structure of the expected reward objective in Equation 4 looks exactly like running policy gradients on a particular stochastic latent-variable policy (where $\pi(a|s; z, [\bar{\theta_k}]\_{k=1}^n)$ = $f_{\sum_{k}z_k\bar{\theta}_k}(a|s))$). With that perspective in mind, I'd be curious to hear the authors intepretation of how the regularization in Eq 5 compares to other works that consider regularized latent variable policies, e.g. (Kumar et al 2020b)?

**Summary Of The Paper:**

This paper considers the problem of learning policies in a training environment that adapts fast to a different test environment. The proposed method learn a subspace of policy parameters that are optimal for the training environment, with the hope that one of these policy parameters will generalize better to the test environment than a single point estimate.

**Summary Of The Review:**

(Copied from above) The problem is interesting, and the proposed method is simple and apparently easy to tune; however, I am concerned about whether or not this method will scale to more challenging environments than the Gym benchmark tasks because of various qualities of the method I describe below. While this is a promising direction of research, I think that this paper requires some iteration before acceptance.

-- Post Rebuttal --

Given the new experiments and analyses, I am increasing my score to a 6.

---

> ### Author Response · Authors · 2021-11-19
> **Answer from authors**
>
> Thank you for your remarks. Please have a look at the “answer to all reviewers”.
>
>
> * **Diversity in behavior/state distribution rather than in parameters:** You are right that our method does not encourage functional diversity at train time as argued in the “answer to all reviewers”. And this is exactly what we expect. Indeed, encouraging functional diversity at train time may prevent the model to find optimal policies over the train environments. Our approach is based on a lighter constraint that is easier to satisfy without hurting train performance (see Figure 3 (left)). This is why, contrary to baselines, the tuning of the beta value is easy with our model, while critical with other approaches (moreover, other methods need to design a good classifier…) Indirectly, our approach encourages functional diversity at test time as it can be observed by the differences between the reward collected by the different policies at test time (see also https://sites.google.com/view/subspace-of-policies/home for videos and the additional experiments on mazes provided in the updated version of the paper - Appendix B.8.). This diversity at test time comes from the fact that, the parameters of the learned policies being different, they process ‘new’ test states differently and thus output different actions. We have added a measure of the functional diversity induced by the different methods in Figure 8 in Appendix and illustration of the learned policies in Figure 4 left.
>
>
> * **Comparison between learning an ensemble of policies vs. best ensemble members in the test environment:** Considering the Single Policy model (and the fact that we averaged results over multiple runs), we did not launch experiments with K randomly initialized policies systematically. We have updated the paper with results where K=5 randomly-initialized policies are trained (see Table 4(top) in Appendix B.1. page 16). As expected, our method is able to discover better test policies through the cosine loss and the sharing of parameters. Note that, training K independent policies when using on-policy algorithms (PPO in our case) needs K times more training samples, K times more parameter storage, and is thus highly inefficient at train time.
>
>
> * **Beta tuning.**  For all the methods, we have trained different models for different values of beta. When not specified, for LoP, we report test results for $\beta=1.0$. For Lc and DIAYN+R, we report test results with the best value of beta (as if we were able to correctly select beta over test environments). This corresponds to an optimistic evaluation of the performance of these baselines that aims at showing that our method is much more efficient since it does not need such a beta-tuning. Said otherwise, we compare our model in a realistic setting ($\beta=1.0$. for all environments) to baseline in an optimistic setting (unrealistic beta fine-tuning) as a way to show the effectiveness of our approach. We make this more clear in the updated version (Section 4).
>
>
> * **Robustness to different levels of variation:** This aspect is studied in Table 7 in Appendix  where we evaluate policies for different levels of input features corruption(Ant environment with simulated “defective sensors” that are zero-ed at test time). As expected, up to a certain variation level (35% of zero-ed input), none of the methods is able to adapt to the test environments, but LoP seems more robust than the other algorithms.
>
>
> * **Structure of the expected reward objective and comparison with SMERL:** The value of z in our model can be seen as a latent variable that models a mixture of policies. It is, in principle, close to methods like SMERL. The first difference is that, in our case, the latent variable z is continuous (while discrete in DIAYN and SMERL). The second one is on the assumption made between the value of this latent variable, and the behavior of the policy. While in the DIAYN/SMERL case, the relation is on the state distribution generated by the policies, it is in the parameter space in our case. Being able to make a link between our method and Bayesian RL is an interesting direction that will be explored as a future work.
>
>
> * **Effective setting of N=2, not the case for N=3:** Learning with high values of n (N>2) may help to discover a larger subspace. But this has two drawbacks: first the learned subspace being larger, the search procedure at test time will take more episodes. This is for instance illustrated in Figure 3 (right) where increasing K when N=3 is providing better test policies, while the performance saturates at K=10 when N=2. Second, since more parameters have to be learned, the training time will be increased (and the need for training memory). For these reasons, we decided to focus on N=2 in the paper.  We report more results on LoP versus CoP in (figure 9 in appendix B1). This will be the focus of future investigations.
>
> Thanks for your remarks, and happy to discuss more if you have other remarks

---

> > ### Comment · Reviewer_UNwx · 2021-11-29
> > **Thank you for the Clarifications**
> >
> > Note: The supplementary website in your response links to a de-anonymized version of your paper.
> >
> > Thank you for the clarifications, additional experiments, and details about hyperparameter configurations.
> >
> > **Diversity in behavior/state distribution rather than in parameters**:
> >
> >      This is why, contrary to baselines, the tuning of the beta value is easy with our model,`
> >
> > I am not convinced about this point (which also concerns the response to **Robustness to different levels of variation**). The hyperparameter $\beta$ controls the effective "diversity" of the learned policy, but does not affect the training time performance. While the authors attempt to cast it as a positive, it seems that this fact means it is actually quite difficult to tune $\beta$ for a new environment, since train-time statistics cannot be used to calibrate how $\beta$ affects this diversity in a new domain.
> >
> > **Comparison between learning an ensemble of policies vs. best ensemble members in the test environment:**
> >
> > Thank you for adding this comparison: it seems based on the scores in Table 4 that the simple ensembling strategy is a very respectable alternative to the compared methods (although certainly less than LoP). Also, while the comment made about sample complexities is true, this can likely be addressed by parameter sharing across ensemble members just as is done for LoP.
> >
> > **New Procgen Results:** Can you comment on Table 20? Why is LoP / DIAYN so much worse than a single policy?

---

> > > ### Author Response · Authors · 2021-11-29
> > > **Follow up**
> > >
> > > Hi, Thanks for the feedback
> > >
> > > With the short rebuttal period, and the many experiments we had to launch, we made two mistakes:
> > > - First, we have updated the website to answer the reviewer qrCv, and switched to the 'full' version of the website with the link to the arxiv version of the paper. We have corrected this. Please, consider that the website was completely anonymous until the rebuttal and this mistake occurred only a few days ago.
> > > - Concerning the procgen table 20,  we have mixed ProcGen results with the TwoRooms Maze results of the previous section.  The correct table is :
> > >
> > > |                   |     Single |        LoP |    DIAYN+R |
> > > |-------------------|-----------:|-----------:|-----------:|
> > > | Levels 100 to 110 | 11.9 ± 2.6 | 20.1 ± 4.4 | 12.1 ± 5.2 |
> > > | Levels 200 to 210 |  7.1 ± 1.7 | 10.3 ± 0.2 |  5.1 ± 1.7 |
> > > | Levels 300 to 310 | 14.3 ± 5.1 | 18.7 ± 2.7 | 17.1 ± 4.6 |
> > >
> > > Many apologies for these mistakes.
> > >
> > > Concerning your point about diversity, we agree that beta in LoP provides less freedom degrees than beta in DIAYN acting as a on/off switch. The good point is that, without no tuning of beta, we obtain good performance. The bad point is that, for particular environments where one needs more diversity, the cosine constraint may be too low. One of our expectations is that we can control the diversity by using more anchor models, (using more anchor models would make the constraint harder to satisfy resulting in an increased diversity), but we have no concrete results and are working on it. We will highlight this aspect more in the final version.
> > >
> > >  Thanks again.

---

> ### Author Response · Authors · 2021-11-25
> **Asnwer**
>
> Hello, we would be grateful if you can confirm whether our response has addressed your comments, and let us know if any issues remain.

---

### Official Review · Reviewer_6h1N · 2021-11-04

**Correctness:** 2
**Technical Novelty And Significance:** 2
**Empirical Novelty And Significance:** 2
**Recommendation:** 3
**Confidence:** 5

**Main Review:**

While I do find the observations interesting, I am not convinced that the method is working for the reasons that the authors have formulated. The training of uniform samples of convex combinations of anchor policies, while regularizing the anchors to be as different as possible, is effectively searching for regions in the policy space that are flat -- the policies parameterized in this way are trained to perform well in expectation. The robust performance should come from the fact that with this region of parameters, the behavior of the system can tolerate small variations better in the first place, regardless of whether it comes from the environment dynamics or the policy parameter changes. One limitation is of course this search is unlikely going to be successful for hard control problems, where not arbitrary perturbations on the policy parameters would work, and in those cases the training itself may in fact perform badly. These problems can be hard to see in the Mujoco environments.

There are a lot of issues of using wrong mathematical terms in the technical part. First of all, the approach is not learning a "subspace" of the parameter space -- in the current formulation it is clearly not closed under linear operations. Also, "simplex" is reserved for the convex hull of n+1 points in an n-dimensional space. So taking a few samples in the policy parameter space, which by itself has a large number of dimensions, the subset described in the paper can not be called "simplex". In section 3.3, it should not be called a "line" which needs to be unbounded in either directions, since it's just a segment.

Overall I think the authors made interesting observations, but the mathematical analysis is weak and does not fully explain the observations.

**Summary Of The Paper:**

The work proposed a method for training robust policies in RL by optimizing convex combinations of different policies, and showed that it achieves good performance against parameter variations in various Mujoco environments.

**Summary Of The Review:**

Overall I think the authors made interesting observations, but the mathematical analysis is weak and does not fully explain the observations.

---

> ### Author Response · Authors · 2021-11-19
> **Answer from authors**
>
> Thanks for your remarks. Please, also consider the “answers to all reviewers” comment.
>
> **[not convinced that the method is working for the reasons that the authors have formulated]** We remind the reviewer that our model aims at learning diversity in the parameter space, instead of functional diversity. However, as mentioned in the “answer to all reviewers”, we have performed additional analysis concerning the model behavior in terms of diversity.
> The new experiments show that our approach induces a functional diversity on the learned policies (see  paragraph “Analysis of the learned policies” in Section 5 and Figure 8 Appendix B.1) which is less strong than the one induced by DIAYN+R approaches. Our model is thus able to find diverse policies without hurting the training performance, finding a good trade-off in the policy space (while DIAYN+R may hurt the training performance if not well tuned). To confirm this, we have launched experiments on a more complex pixel-based environment (ProcGen -- see Table 1 and Appendix B.7) and on a simple one where it is possible to visualize the learned trajectories (Maze 2d -- Appendix B.8). We expect that these new experiments bring a better understanding of how our method is working.
>
> **[Mathematical terms used in the paper]** We want to first point out that, since our method is built on [Wortsman et al. 2021], we have decided to keep the same terminology than the one in this paper to make the connection between our work and the previously proposed model.
> Concerning the ‘subspace’ term, it can be interpreted in its general sense: ‘a space that is wholly contained in another space, or whose points or elements are all in another space’ and our method is indeed learning a subspace of policies i.e., an infinite subset of policies that is included in the set of all possible policies (given a particular policy architecture).
> Concerning the use of the term “simplex”, we recall the definition of a simplex here: ‘a k-simplex is a k-dimensional polytope which is the convex hull of its k + 1 vertices’. Given $N$ anchor policies in the space of hyperparameters $\mathbb{R}^d$ ($N << d$) we indeed aim to find a $N-1$ simplex in $\mathbb{R}^d$. The fact that the N anchor policies are affinely independent during learning is guaranteed by the cosine similarity penalty term. The line segment is the most obvious and easiest simplex to learn (it is a 1-simplex in R^d). We have updated the paper to make it clear that the LoP model corresponds to a $N-1$ simplex.
> Concerning the term ‘line’, we use the term introduced in [Wortsman et al. 2021]. It is already explained in Section 3.3 that “In the case of N=2, the subspace of policies corresponds to a simple segment in the parameter space”.
>
> **[“These problems can be hard to see in the Mujoco environments: ”]** MUJOCO environments are widely used environments in RL that are known to be quite realistic. Having a method working well on this family of problems is, in our opinion, a strong result, particularly knowing that we proposed results in 6 different environments and different variations of these environments, leading to 50 test environments. We also proposed results on gym minigrid and other simple gym settings in the original submission. We have now added results over the ProcGen environment to explore a different setting (i.e pixel-based) (Table 1 and Appendix B.7) and on a maze 2d (Figure 4(left) and Appendix B.8) to illustrate the learned policies. Note that the experiments are very large in terms of GPU consumption, and having more experiments than what we propose is extremely costly and difficult to achieve.

---

> ### Author Response · Authors · 2021-11-25
> **Answer**
>
> Hello, we would be grateful if you can confirm whether our response has addressed your comments, and let us know if any issues remain.

---

### Author Response · Authors · 2021-11-19
**Answer to all reviewers**

We would like to thank reviewers for their careful reading and all their insights. We also provide individual responses for each reviewer.

We would like to remind reviewers that our article is an algorithmic paper proposing a new algorithm to handle online adaptation in RL. Our method LoP aims at injecting diversity in the parameter space instead of diversity in the behaviors/state distribution; The latter diversity is also called functional diversity and is the diversity proposed in [Kumar et al., 2020b; Osa et al., 2021].  As an output, we show that our model is able to obtain better online adaptation performance, with less need of parameter tuning, resulting in a realistic approach to online adaptation. Qualitative results of the k-shot evaluation can be visualized at https://sites.google.com/view/subspace-of-policies/home

All the modifications and supplementary materials are highlighted in orange in the updated version.



### Qualitative analysis
* **Functional diversity generated by LoP** (new experiments - Section 5, Figure 4, and Section B.8 in appendix): To better understand the nature of the policies learned by LoP, we have evaluated the functional diversity of the policies. This diversity is measured by the accuracy of a discriminator learned a posteriori to discriminate the different policies (i.e. z values) based on the sampled states (Figure 8 in Appendix B1). Logically, DIAYN+reward exhibits a high ‘functional’ diversity at train and test times (accuracy is 100%). Note that this diversity may hurt performance at train time (Figure 10 in Appendix B2).  The policies obtained by LoP are also diverse on the train environment (accuracy is about 82 %), but interestingly more diverse on test environments (accuracy is 87 %): as expected the diversity in the parameter space makes the different policies to react differently to unseen states (at test time), generating more diversity at test time. Note that LoP does not hurt the train performance.  We also propose visualisation of trajectories in a new  2d goal maze setting (Figure 4 (left), Appendix B.8) for a better understanding of the behavior of the different methods.


* **Policies usage at test time:** We plotted histograms illustrating which policy is used on each test environment with LoP and DIAYN+R (see Figure 4 (right) in the main paper and Figure 7 in Appendix B.1.). This confirms that i) the anchor policies are not always the one used at test time showing the importance of learning a complete subset ii) different policies are used for different test environments showing the variety of policies learned by LoP.


* **Benefits of using N=3 when K is high:** We plotted the evolution of the k-shot evaluation reward with respect to K, from K=1 to 100 in Figure 9 (Appendix B1).


* **Sensitivity to corrupted input features:** (already presented in the original paper, but only in appendix, now in the main paper Section 5).  We analyze the ability of the different methods to handle corrupted input features. This is typically one of the settings on which we expect LoP to work well since, through the diversity in the parameter space, it will learn policies that ‘weight’ the input features differently, and that will react differently to corrupted values. Our results on the MUJOCO Ant environment confirm this intuition.



### Baselines and Complex Environments
* **Baseline - Ensemble of policies:** We have computed the performance of a simple set of K=5 policies trained independently. Note that such a method needs K times more samples than the ‘Single Policy’ baseline and is unrealistic with large values of K (see Appendix B.1 - Table 4 and comments in Section 4). As expected, without any constraints, the set of independent policies is not able to perform as well as LoP to unseen environments.


* **Baseline - Kumar et al. 2020:** We detail in Section 4 why we do not compare with [Kumar et al. 2020] following the observations already made by the authors of [Osa et al. 2021].


* **More complex environments:** We want to recall that our article proposes results on many different environments including classical control tasks (e.g. mujoco), but also a maze problem (gym minigrid) for both discrete and continuous action spaces.  We have added a new experimental campaign on pixel-based RL on the ProcGen environment (see Section 5 and Table 1 - more games will be included upon acceptance - each experiment on ProcGen is about 12 GPU.hours and it is not possible to make more experiments upon rebuttal ending period).  Again, LoP outperforms the different baselines. We have also new experiments in a toy maze2d problem (Figure 4-left and appendix B.8) Please, consider that the experiments in the article correspond to thousands of GPU.hours and it is difficult to do more. We expect to now cover a larger set of types of environments

* **Writing:** We have clarified the notations and vocabulary used in the paper following reviewers suggestions.

---

### Decision · Program_Chairs · 2022-01-20

**Decision:**

Accept (Poster)

**Comment:**

The paper tackles the problem of generalizing to a new environment by learning a small set up anchor policies (even just 2 for the final approach) which span a sub-space that can be searched efficiently in a new environment.
The discussion and additional experiments managed to convince most reviewers that the method indeed works as the authors had hypothesized (especially regarding functional diversity). At the moment the analysis is mainly based on empirical observations, it would be good to also have a thorough theoretical analysis of the method.